# IS IN-CONTEXT LEARNING LEARNING?

**Adrian de Wynter**
Microsoft and the University of York
adewynter@microsoft.com

## ABSTRACT

In-context learning (ICL) allows some autoregressive models to solve tasks via next-token prediction and without needing further training. This has led to claims about these model's ability to solve (learn) unseen tasks with only a few shots (exemplars) in the prompt. However, deduction does not always imply learning, as ICL does not explicitly encode a given observation. Instead, the models rely on their prior knowledge and the exemplars given, if any. We argue that, mathematically, ICL fits the definition of learning; however, its full characterisation requires empirical work. We then carry out a large-scale analysis of ICL ablating out or accounting for memorisation, pretraining, distributional shifts, and prompting style and phrasing. We find that, empirically, ICL is limited in its ability to learn and generalise to unseen tasks. Namely, in the limit where exemplars become more numerous, accuracy is insensitive to exemplar distribution, model, prompt style, and the input's linguistic features. Instead, it deduces patterns from regularities in the prompt, which leads to distributional sensitivity, especially in prompting styles such as chain-of-thought. Given the varied accuracies and on formally similar tasks, we conclude that autoregression's *ad-hoc* encoding is not a robust mechanism for learning, and suggests limited all-purpose generalisability.

## 1 INTRODUCTION

In learning theory, learning is tantamount to generalisation. Learning how to solve a task means that, after seeing examples of a task drawn with a distribution $\mathcal{P}$, a learner will have a bounded probability of error on classifying new inputs from some $\mathcal{Q} \neq \mathcal{P}$ (Valiant, 1984). In most traditional learning paradigms, a learner observes inputs from $\mathcal{P}$, and then encodes them within its own knowledge (e.g., updating weights). Then it uses this knowledge to generalise to new examples. Autoregressive large language models (LLMs)[1] do not *explicitly* do that. Instead, they perform *in-context learning* (ICL), where they 'solve' (produce relevant outputs for) a task specified in natural language via next-token prediction (Brown et al., 2020). An LLM observes, but does not encode, the full training set through the prompt. Instead, it updates its beliefs *ad hoc*, where the beliefs are a combination of the input (drawn from $\mathcal{P}$) plus its own intrinsic knowledge (frozen weights). That is, it modifies its states at run-time to–ideally–generalise to new observations for *any* $\mathcal{Q}$. Reliance on intrinsic knowledge implies that the LLM is also expected to generalise to *any task* (unseen $\mathcal{P}$).

We argue, however, that **knowing is not always the same as learning**. Claims on an LLM's *in-task* generalisation (consistent performance w.r.t. any $\mathcal{P}$ within the task) and *cross-task* generalisation (consistent performance w.r.t any task) have divided the field. Theoretical characterisations on their learning power are usually limited, and hence Borenstein et al. (2024) argued that empirical explorations could help understand what *can* transformer-based models learn versus what they *actually* learn–a central motivation of our work. However, criticisms to empirical research on LLMs note that prompt-and-model dependence makes it hard to reproduce (Li et al., 2025; De Wynter, 2025; Sclar et al., 2024), and that the terminology, methods, and results themselves are unreliable and could lead to misinterpretation (Huang & Chang 2023; De Wynter 2025; also Section 2).

In this paper we examine to what extent ICL is an effective learning paradigm. We begin by noting that, mathematically, ICL constitutes learning–as opposed to solely repeating internal knowledge–

---

[1]We use 'autoregressive model' and 'LLM' interchangeably, with the assumption that the LLMs discussed are autoregressive.

but remark that further work is required to fully characterise it. We then perform empirical studies accounting for some of the criticisms and shortcomings mentioned; namely, sensitivity to pretraining, memorisation, and dependence on natural language; prompting style and phrasing; and robustness to distributional shifts in the observed training and test sets. Thus we focus on evaluating generalisation from data *in context* (unseen $\mathcal{P}$ until runtime, and fully-unseen $\mathcal{Q}$); as opposed to generalisation from a model's pretraining strategy.

Our experiments are on four LLMs, nine tasks, and ablations on prompting strategies, training distributions, and exemplar setups. The main results–that is, test set results–comprise 1.89M predictions per LLM. To our knowledge, our study is one of the largest of its kind.[2]

## 1.1 FINDINGS

We ablate dependence on natural language and prompt phrasing, and use artificial alphabets to force LLMs to learn the task solely from the observations. Hence, our findings seek to characterise ICL as a learning paradigm, and not as an evaluation of prompt-based problem-solving capabilities. These are:

1. In the limit (as the number of exemplars grows), the average accuracy gap narrows between the LLMs tested, and all prompting strategies steadily improve accuracy. Likewise, semantically nonsensical prompts near or match their non-randomised counterparts.

2. ICL under altered training (exemplar) distributions is robust to positionality and proportion of labels in the limit. However, ICL is brittle to altered test distributions (i.e., out-of-distribution; OOD) as the distance between train and test distributions grows, especially in chain-of-thought (CoT; Wei et al. 2022b) and automated prompt optimisation (APO).

3. Closely-related tasks do not necessarily have closely-related performances, with average accuracy differences as large as 31%. Moreover, traditional baselines (e.g., decision trees and kNNs) outperform ICL average performance in half of the tasks evaluated.

Our findings contradict the notion that a few exemplars are required to solve a task: peak average accuracy was at 50-100 exemplars–much higher than in reported negative (Lu et al., 2024; Sclar et al., 2024; Dziri et al., 2023; Delétang et al., 2023) or positive (Brown et al., 2020) results from LLM studies in natural-language and automata-based tasks, and independently confirms similar results for natural-language tasks (Anil et al., 2022; Agarwal et al., 2024). They, however, counter the view that LLMs are brittle to exemplar ordering or characterisation (Sclar et al., 2024; Errica et al., 2025; Zhao et al., 2025; Agarwal et al., 2024), and align with the view that CoT and APO are good at solving some tasks (Merrill & Sabharwal, 2024; De Wynter et al., 2023a; Li et al., 2024), although we show that these are not robust to OOD. Finding 3 expands on the works that found that LLM accuracy decays with task complexity (Dziri et al., 2023; Gupta et al., 2025; Merrill & Sabharwal, 2024), but notes that analogous tasks have marked performance differences.

## 1.2 INTERPRETATION

Our findings are constrained to easily-verifiable tasks (e.g., parity checking or Hamiltonian-cycle verification) in a single call. From within our theoretical framework, we find evidence that ICL presents signs of learning capabilities; but that it is tied to the autoregressive paradigm, and not to any particular LLM, training strategy, or prompting style. We argue that this is because **ICL leverages statistical features from the prompt, as opposed to feature relations within the data**. This *ad hoc* encoding mechanism implies that ICL's **cross-task generalisability is limited** to the representativeness of the data. Thus, we conclude that, from the perspective of our framework, ICL is mathematically a form of learning, albeit not a robust one.

Remark that our work is constrained to **non-natural language tasks**, with our ablation on natural language limited to the instructions, not the datapoints. This setup forces LLMs to not rely on their intrinsic knowledge, and instead infer the full characterisation of the data from the prompt itself. This suggests differences with some claims on their emergent capabilities, but also raises the question to which semantic extent priors from the data impact (realistic) natural language tasks and its relationship to learning. This opens avenues for further systematic evaluation of their capabilities.

---

[2]Code and data is in `https://github.com/adewynter/is-icl-learning`

## 2 RELATED WORK

Evaluation of LLMs is an active area of research, and our coverage of its works is non-exhaustive. For broader surveys of this area we point the reader to Huang & Chang (2023), Li et al. (2025), and Qiao et al. (2023). For ICL in particular, see Zhou et al. (2024). Early research focused on evaluating whether RNNs, transformers, and other non-generative models actually performed learning (Borenstein et al., 2024; Zhang et al., 2023; Butoi et al., 2025). These works, like ours, investigated the models' ability to learn formal languages, or subsets of first-order logic, and also found brittleness in OOD scenarios. A solution proposed by Dan et al. (2022) involved passing in the encoding of the automaton generating the language–we explore its viability for ICL in our work.

### 2.1 THEORETICAL EVALUATIONS

Theoretical research on what transformer-based models can possibly learn often find negative results (Hahn & Rofin, 2024; Strobl et al., 2024a; Kleinberg & Mullainathan, 2024). Even when it has been known for some time that the transformer (under certain assumptions) is Turing-complete Pérez et al. (2021); Bhattamishra et al. (2020); Li & Wang (2025), Turing completeness by definition requires an unbounded resource or finding an appropriate machine; which is itself undecidable, although approximable (Wei et al., 2022a). More constrained works with specific attention types could *recognise* languages in the class of constant-depth, polynomial-size alternating circuits. Concretely, those in $AC^0$ (Hao et al., 2022); and partly $TC^0$ ((Strobl, 2023); (Li et al., 2024) for CoT), although as of yet it is unknown why (Strobl et al., 2024b). Nonetheless, Kleinberg & Mullainathan (2024) showed that next-token prediction is a different problem than judging membership (labelling data). Even their ability to model formal languages tends to find disparate results, depending on the assumption made (Strobl et al., 2024b). It is perhaps because of these findings that Borenstein et al. (2024) call for an empirical evaluation of effective capabilities of LLMs.

### 2.2 EMPIRICAL EVALUATIONS

Empirical LLM evaluation is complex and also marred with disparate accounts on their capabilities. This is often due to the influence of various factors, ranging from choice of model, statistical significance, or ablations with respect to natural language and memorisation (De Wynter, 2025). For example, it is known that several LLMs suffer from data contamination (Carlini et al., 2023; Lee et al., 2023; De Wynter et al., 2023b) which could render benchmark-based evaluation unreliable; and that different measurements show less impressive results (Schaeffer et al., 2023; Altmeyer et al., 2024). Likewise, Gupta et al. (2025); Dziri et al. (2023); Merrill & Sabharwal (2024); Liu et al. (2023) and Lu et al. (2024) evaluated (and found weaknesses) in LLMs when generalising to unseen tasks, especially when using CoT and as the task complexity grew. On the other hand, positive results such as that of Ontanon et al. (2022) and Borenstein et al. (2024) indicate that, for certain tasks, these weaknesses may not necessarily hold. Indeed, some positive results, such as that of Agarwal et al. (2024), showed that expanding shots improved performance in natural and non-natural language problems, albeit the main results were constrained to a single model.

Research has also attempted to determine whether the models *understand* the task as described by the prompt, usually with negative results (Webson & Pavlick, 2022; Jang et al., 2023; De Wynter & Yuan, 2025; Mancoridis et al., 2025; Dziri et al., 2023; Strobl et al., 2024a) Proposed explanations to this related model size to sensitivity to semantics (Shivagunde et al., 2024; Long et al., 2024) and inductive/selection biases Zhao et al. (2025); Chang & Bisk (2025) although this sensitivity disappeared when the exemplars included instructions.

However, there were some–reasonable, due to scope–gaps in the works above due to the limiting factors mentioned. Thus, we attempt to account for these in our work. Other attempts to explain ICL have been through benchmarks (Yauney & Mimno, 2024; Mirzadeh et al., 2025; Zhuo et al., 2024; Sclar et al., 2024), mechanistic interpretations (e.g., subnetwork generalisation (Bhaskar et al., 2024; Kumon & Yanaka, 2025; Hu et al., 2025); probing (Yin & Steinhardt, 2025; Azaria & Mitchell, 2023; Todd et al., 2024; Ju et al., 2024)), Bayesian approaches (Xie et al., 2022; Edelman et al., 2024), or more targeted evaluations, such as that of Chan et al. (2022). This latter work argues that ICL arises from the distribution of the elements within the training data, along with the use of the transformer, and it is a driving argument for our work.

## 3 BACKGROUND: THE NEED FOR EMPIRICAL EVALUATION OF ICL

We discuss formalisms for learning and task similarity in Sections 3.1 and 3.2, and tie ICL to these in Section 3.3, noting that they partly overlook the mechanism behind ICL. Details are in Appendix B.

### 3.1 A FORMAL DEFINITION OF LEARNING

We capture robustness in learning with a variation of the probably approximately correct (PAC) framework from Valiant (1984). We use PAC learning as it is the predominant model in computational learning theory–concretely, statistical learning theory–as well as in language acquisition (Mitkov, 2022; Niyogi, 2006). It also allows for some leeway to a learner through error tolerance. For a comparison with other frameworks, see Appendix C.

We reframe PAC learning to focus on the learner. This is a syntactical re-definition and does not alter the original framework. Suppose we wish to model a binary classification task with features assumed to be drawn from some nonempty set $X \subset \mathbb{R}^m$. These examples are labelled with an unknown function $c \colon X \to \{0, 1\}$. In machine learning, a (data)set $D$ is sampled with some distribution $\mathcal{P}$ supported on $X$, $D = \{\langle x_i, c(x_i) \rangle | x_i \sim \mathcal{P}\}$. A learner (algorithm) $f \colon X \to \{0, 1\}$ observes $D$ until its empirical error $error(\cdot)$ is bounded by some $\epsilon \in (0, 1/2)$, where

$$error(f, D) = \frac{1}{|D|} \sum_{\langle x, c(x) \rangle \, ; \, x \in D} \mathbb{1}[f(x) \neq c(x)] \leq \epsilon. \tag{1}$$

Equation 1 must holds for any other dataset $E$ and distribution $\mathcal{Q}$ such that $E = \{\langle x_i, c(x_i) \rangle | x_i \sim \mathcal{Q}\}$, where $\mathcal{Q}$ is likewise supported on $X$; that is, if

$$\Pr[error(f, E)] \geq 1 - \delta, \tag{2}$$

for $\delta \in (0, 1/2)$. Intuitively, a learner has learnt the task if it has a (lower) bound on its error for any datapoint. Since $\mathcal{P}$ and $\mathcal{Q}$ are unspecified, $f$ **has learnt** $c$ if it is *robust* to changes in $\mathcal{P}$.[3] Standard PAC learning has some limitations, especially around regular languages. Hence, our reframing is only to ground our discussion on a strict definition of learning, as done by, e.g., Livni et al. (2014).

### 3.2 TASK SIMILARITY

In formal language theory, a collection of transition rules $\mathcal{G}$ (a grammar) generates instances (strings) using symbols from an alphabet $\Sigma$ to form a language $L$. We assume all instances of a task are generated by its own $\mathcal{G}$ and $\Sigma$, with transition probabilities given by a (chosen) $\mathcal{P}$. This $\mathcal{P}$ is the same from Section 3.1, and, to the learner $L$ may be known (or deduced), but $\mathcal{G}$ is not. Formal languages may be classified according to the (expressive) power of the automaton able to accept/reject (recognise) an $x$ based on the query $x \in L$?. Relevant to us are these recognisable by finite state automata (FSA), and pushdown automata (PDA). FSA read the input unidirectionally, changing their internal state between accept and reject, and return either when finished. Tasks such as PARITY and Hamiltonian-cycle verification, are recognisable with an FSA. Other tasks, like stack manipulation, require the automaton to track a set of states. PDA are FSA equipped with memory, and can solve these, more complex, tasks. They are considered more powerful than FSA. The autoregressive nature of LLMs allows for some memory, and hence they could be considered a type of PDA. However, in this work we treat LLMs as recognisers of unknown expressive power.

### 3.3 DEFINING ICL IN CONTEXT

In ICL, an LLM takes in a natural-language string as a task specification (system prompt), and uses the input tokens (in natural language) to 'solve' (learn) it by predicting the following tokens recursively. Formally, Wang et al. (2023) formulate ICL classification as

$$\underset{f(x_k) \in \{0,1\}}{\operatorname{argmax}} \ \Pr[f(x_k) = c(x_k) | x_1, c(x_1), \dots, x_k], \tag{3}$$

---

[3]In the words of Niyogi (2006), any classifier 'worth their salt' should fulfil this condition.

where $x_k$ is the datapoint to be labelled, and we have used the notation from Section 3.1. However, ICL is sensitive to the system prompt. Thus, practitioners have resorted to various prompting techniques, which are not accounted for in Equation 3. Factoring in both we get

$$\underset{f(x_k)\in\{0,1\}}{\text{argmax}} \Pr[f(x_k) = c(x_k)|p, \pi(x_1), \pi(x_2), \ldots, \pi(x_{k-1}), \tilde{\pi}(x_k)], \quad (4)$$

where $p$ is a system prompt, $x_i$ are example datapoints ($i < k$), and $x_k$ is the instance to be classified. We let $\pi, \tilde{\pi}$ be functions that take in inputs $x_i$ and return natural-language representations $\pi(x_i) = \langle x_i, c(x_i)\rangle$ for $i < k$ (r. $\tilde{\pi}(x_k) = x_k$). These could be, e.g., a concatenation of the datapoint and the label (e.g., $\pi(x_i) =$ '$x_i : c(x_i)$'); and a datapoint conditioning for next-token (label) prediction, $\tilde{\pi}(x_k) =$ '$x_k$ :'. It could also be more complex (e.g., 'Let's think and solve step-by-step...'). Both $p$ and $\pi(x_i)$ may be empty, but not at the same time. At inference time, when computing $c(x_k)$, the LLM conditions recursively on its observations from $p, \ldots, \tilde{\pi}(x_k)$, and its previous knowledge.

PAC learning does not limit *how* the learner learns. From Equation 4, it follows that ICL can be viewed as a (formal) learning process. Namely, a learner $f : \{p\} \times_k X \to \{0, 1\}$ is an LLM with $k - 1$ exemplars $x_1, \ldots, x_{k-1} \sim \mathcal{P}$, an input instance to classify $x_k \sim \mathcal{Q}$, representations $\pi, \tilde{\pi}$ and an optional system prompt $p$. We say that ICL learns $c$ and $X$ if Equation 2 holds for any $x_k, x_1, \ldots, x_{k-1} \in X, \pi, \tilde{\pi}, \mathcal{P}$, and $\mathcal{Q}$. This thus makes ICL strongly dependent on $\pi, \tilde{\pi}$ and $p$ (the prompt), but *does not specify* to what extent, since it depends on the autoregressive nature of the LLM (namely, the 'scratchpad') and its own weights. Indeed, one consequence of Equation 4 is that as $k$ grows, since $p$ is constant, its contribution vanishes when equiprobable to the exemplars:

$$\Pr[Y|p, \pi_1, \ldots, \pi_{k-2}, \tilde{\pi}_k] \propto \Pr[p|Y] \left(\prod_i^{k-1} \Pr[\pi_i|Y]\right) \Pr[Y] \Pr[\tilde{\pi}_k], \quad (5)$$

where we let $Y := f(x_k) = c(x_k)$; $\pi(x_i) := \pi_i$; and $\tilde{\pi}(x_i) := \tilde{\pi}_i$, for readability. Conversely, the encoded exemplars have a major contribution in the limit. Nonetheless, our reframings do not characterise $\pi$ (e.g., which natural-language strings, if any, work better?). This thus calls for empirical evaluations as to *how* effective ICL is at learning, accounting for $\mathcal{P}, p, \pi, \tilde{\pi}$, and $c$.

## 4 METHODS

Sample prompts are in Appendix H and full task definition and characterisations with respect to ID/OOD are in Appendix E. Specifics on LLM calls are in Appendix F.

### 4.1 FRAMING

We seek to find out if a learner (LLM) $f$ can correctly and robustly decide if a given $x \in \Sigma$, sampled with some $\mathcal{D}$ for some $\Sigma$ and $\mathcal{G}$, belongs to a language $L$. We let $\mathcal{G}, \Sigma$, and $L$ be fixed for a task, but not always known to $f$. We measure correctness with accuracy, $1 - error(f, \cdot)$; and robustness with accuracy under the **distributional shift**. That is, we consider both in-distribution (ID) entries $x \sim \mathcal{P}$ and OOD entries $x \sim \mathcal{Q}$, for select values of $\delta = ||\mathcal{P} - \mathcal{Q}||_\infty$.

### 4.2 PROMPTING STRATEGIES AND SCOPE

We test prompts that perform a single call to the LLM. More complex strategies, such as Tree-of-Thoughts (Yao et al., 2023) have good performance, but rely on multiple model calls per instance, and hence are not in the scope of our work. We also consider only *single* next-token prediction, as well as robustness to system prompts (i.e., CoT and APO). Reasoning models like o3-mini (OpenAI, 2025), which have a baked-in non-controllable CoT, are thus not in scope. The prompts tested are:

**n-Shot Learning:** Provide $n$ exemplars of an input $x$ and desired, formatted output $\tilde{\pi}(x)$. When we do not provide a system prompt, we refer to it as **Modus Ponens**.

**Description:** Add in the system prompt $p$. This is the usual way to prompt LLMs.

**APO:** A meta-prompting ('prompting to prompt') approach where the LLM adapts its own system prompt $p$ with a development set. It has been shown that this strategy yields better perceived results

than description (De Wynter et al., 2023a). We used the algorithm from Pryzant et al. (2023) to generate $p$.

**Direct Encoding (DE):** Pass in the system prompt plus $\mathcal{G}$ and $L$. This is common in theoretical computer science; in addition, LLMs have been claimed to be capable of understanding code. Note that DE is known to increase robustness to OOD in LSTMs and RNNs (Dan et al., 2022).

**Chain-of-Thought (CoT):** Generate a series of steps leading to the desired output with a predefined scheme in the system prompt.

**Word Salad:** Replace the natural strings from the description with random words. When we apply word salad to the CoT prompt, we call it **Salad-of-Thought** (SoT).

These strategies may be mixed. For example, CoT with word salad and 5 exemplars is 5-shot SoT. Word salad and SoT are considered only in Section 6.1. All prompts were ran with 0, 2, 5, 10, 20, 50, and 100 exemplars; except modus ponens (no zero-shot), and CoT/SoT (no 2-shot due to cost). All prompts had output format specifications (implicit in modus ponens) to facilitate parsing.

### 4.3 TASKS OVERVIEW

All tasks have their own $\Sigma$, and were selected for being closely-related tasks often seen in LLM evaluations, or well-known problems in computer science. All (except one) are decision problems to fit the model from Section 3. We discuss this further in Appendix E.

**PARITY**: (FSA) decide if a given binary string has even zeros. Also known as the XOR function.

**Pattern Matching** (FSA): decide if $abcabb$ is a substring of a given string $x \subset \{a, b, c\}^*$.

**Reversal** (PDA): given a string $l\#r \subset \Sigma$, decide if $l$ equals the reversed $r$, $l = r^{-1}$.

**Stack** (PDA): given final and initial strings $s_f, s_0 \subset \Sigma$ and a series of operations $Op$, decide if $s_f = Op(s_0)$. The operations simulate a stack (push/stop/pop) and may or may not be grammatical (e.g., stack overflows).

**Hamiltonian** (FSA): given a graph in adjacency matrix form and a path, decide if it is Hamiltonian.

**Maze (Complete)** (FSA): given a maze, two segments of a path, and a sequence of moves, decide if the moves connect both segments. Segment separation is never more than three moves.

**Maze (Solve)** (FSA): given a maze and a sequence of moves, decide if the moves form a valid path from start to exit.

**Vending Machine (Verification)** (FSA): given a list of items and costs $C$, a sequence of operations $Op$ (add balance, purchase item), and initial and final balances $b_0, b_f$, verify if $b_f = Op(C) + b_0$.

**Vending Machine (Sum)**: Same as the verification version, but the learner must compute $b_f = Op(C) + b_0$ for an unknown $b_f$. It has a constrained set of moves, but unbounded states ($\mathbb{N}$). This is the only task in our work that is not a decision problem.

### 4.4 MODELS AND MEASUREMENT

We tested four LLMs: GPT-4 Turbo (Open AI, 2023), GPT-4o (OpenAI, 2024), Mixtral 8x7B instruct v01 (Jiang et al., 2024), and Phi-3.5 MoE Instruct (Bilenko, 2024). We measure performance with accuracy, and report standard deviation ($\sigma$) to indicate variation over an average. We use ordinary least squares (OLS) to measure changes. We set all outputs non-compliant with the requested format as zero, but revisit this in Section 6.4. When reporting aggregate numbers, however, we do not factor in out-of-token errors. For baselines, we tested decision trees (DT), k-nearest neighbours (kNN), and a multilayer perceptron (MLP) in succession, and reported the best. We did not baseline path-based problems or arithmetic, since they are often solved with heuristics (e.g., A*).

### 4.5 DATA GENERATION

We created datasets per-task with automata with state transition probabilities drawn from a chosen $\mathcal{D}$. They are synthetic to account for (a) the task's $\mathcal{G}$ and $\Sigma$; and (b) ID and OOD. Every task has

different manifestations of OOD (e.g. the size of a maze). See Figure 1 for a sample automaton and Appendix E for full description of the characterisation of OOD per task.

Figure 1: Data generator for PARITY. Each state has transition probabilities $\delta$, and an emission probability. There is a symmetric automaton with emissions at **0**.

All entries relied on natural language as little as possible (e.g, binary strings or arbitrary symbols in $\Sigma$, such as $^-\backslash\_(ツ)\_/^-$). The training dataset was 2000 entries drawn from a $\mathcal{P}$, and we also generated five balanced, deduplicated test sets, each from a $\mathcal{Q}$ such that $||\mathcal{P} - \mathcal{Q}||_\infty = \delta$, for $\delta \in \{0, 0.2, 0.45, 0.65, 0.85\}$, where $\delta = 0$ is ID, and the rest OOD. This allowed to measure the separation between distributions and the gradual change in performance.

Every test set is 2000 entries, but due to cost we only evaluated 1000. We also mislabelled entries w.p. $\eta = 0.05$ to account for any potential memorisation. Hence, the maximum accuracy for any $f$ that actually learns the task is $95\%$. We only use the training set for APO and the selected baselines. The full suite is 1.89M datapoints.

## 5 RESULTS

We provide results on our analysis: general accuracy (Section 5.1); distributional shifts (Section 5.2); and fine-grained analysis (Section 5.3). For detailed results, see Appendix G.

### 5.1 OVERALL PERFORMANCE

The best average accuracies, per LLM, were in Pattern Matching ($94\pm1\%$; solved the task), Hamiltonian ($85\pm4\%$), and Vending Machine (Verification; $83\pm9\%$). Best accuracies in the worst-performing tasks were Vending Machine (Sum; $16\pm1\%$), Reversal ($61\pm11\%$), and Maze Solve ($63\pm13$). See Table 1 for best-of and averaged results per-model over tasks; and Table 2 for averaged result data per-task over prompts. **LLMs outperformed traditional baselines** (e.g., kNN) **in best-of, but not average best**, scenarios in all tasks except PARITY.

**The best prompt per problem was CoT**, in four tasks. The worst prompt was 2-shot modus ponens, in five tasks. The tasks where CoT underperformed (Pattern Matching and Hamiltonian) were not the same where it was the best-performer. In Maze Complete, however, modus ponens was the worst (2-shot; $9\pm16$) and the best (100-shot; $77\pm5$). Without Vending Machine (Sum), the only non-classification task in our work, the accuracies numbers increased by $5\pm1$ on average.

**Better performances were given by more shots, on average**, when looking at the slope from OLS fits between per-model averages over shots (Table 2). Larger slopes (trends in accuracy improvements) were in modus ponens (8.3) and lowest in CoT (3.3). Mixtral improved the most with more shots, with an average slope of 7.3 (versus 5.8, 3.5, and 4.0 for Turbo, GPT-4o and Phi-3.5).

| Task | Turbo | GPT-4o | Phi-3.5 | Mixtral | Average (Best) | | Average (Worst) | | ML | |
|------|-------|--------|---------|---------|----------------|--|-----------------|--|----|--|
| PARITY | 76 | 90 | 83 | 83 | $80\pm3$ | 100-APO | $16\pm20$ | 2-m.p. | 95 | MLP |
| P. Match. | 96 | 95 | 95 | 95 | $94\pm1$ | 50-DE | $24\pm20$ | 5-CoT | 87 | kNN |
| Reversal | 71 | 77 | 54 | 55 | $61\pm11^*$ | 100-CoT | $20\pm21$ | 2-m.p. | 72 | kNN |
| Stack | 86 | 92 | 66 | 76 | $73\pm14^*$ | 50-CoT | $20\pm21$ | 2-m.p. | 72 | kNN |
| V.M. (Ver.) | 94 | 90 | 84 | 78 | $81\pm12$ | 10-CoT | $22\pm22$ | 2-m.p. | 84 | DT |
| Maze (Comp.) | 83 | 72 | 79 | 81 | $77\pm5$ | 100-m.p. | $9\pm16$ | 2-m.p. | – | – |
| Maze (Solve) | 70 | 61 | 66 | 60 | $63\pm5$ | 50-desc. | $17\pm20$ | 0-APO | – | – |
| Hamiltonian | 93 | 92 | 86 | 85 | $89\pm2^*$ | 100-desc | $29\pm8$ | 0-CoT | – | – |
| V.M. (Sum) | 18 | 20 | 15 | 20 | $16\pm1$ | 5-CoT | $0^\dagger$ | 0-DE | – | – |

Table 1: Maximum accuracies per-model per-problem and peak averages (per shots, over models). An $^*$ is an average over fewer models due to out-of-token failures; a $^\dagger$ means a tie. The best prompts often included natural-language descriptions (CoT, APO, Description). The worst prompt was often 2-shot modus ponens: it lacks a description and led to parsing errors in few-shot. Closely-related tasks had differences of up to $31\%$ accuracy. All baselines degraded in OOD except in PARITY.

| | Prompt | Turbo | | GPT-4o | | Phi-3.5 | | Mixtral | | Avg. slope for acc. |
|---|---|---|---|---|---|---|---|---|---|---|
| | | Slope | Acc. | | | | | | | |
| **Shots** | Modus Ponens | 12.8 | 28±23 | 11.4 | 43±20 | 5.2 | 50 ± 9 | 3.9 | 50±9 | 8.3±3.9 |
| | Description | 3.4 | 57±6 | 1.4 | 56±3 | 4.4 | 50±9 | 8.2 | 48±19 | 4.4 ± 2.2 |
| | DE | 3.0 | 54±5 | 1.4 | 58±3 | 5.5 | 50±10 | 8.1 | 48±20 | 4.5±2.4 |
| | **Word Salad** | 9.8 | 32±18 | 12.1 | 43±22 | 11.5 | 39±21 | 9.8 | 44±20 | 11±4.6 |
| | APO | 6.1 | 51±11 | 2.0 | 57±4 | 4.6 | 51±9 | 8.4 | 48±1 | 5.4± 2.6 |
| | CoT | 3.4 | 47±6 | 1.3 | 55±4 | 0.5 | 45±1 | 8.0 | 38±15 | 3.3±2.4 |
| | **SoT** | 1.8 | 20±4 | 3.5 | 25±7 | 0.3 | 26±4 | 1.8 | 22±5 | 1.6±2.2 |
| **OOD** | Modus Ponens | -0.3 | 28±1 | -0.6 | 43±1 | -0.6 | 50±1 | -0.2 | 50±1 | -0.4 ± 0.4 |
| | Description | -0.5 | 57±1 | -0.8 | 56±1 | -0.5 | 50±1 | -0.5 | 48 | -0.5 ± 0.4 |
| | DE | -0.4 | 54±1 | -0.9 | 58±1 | -0.4 | 50±1 | -0.1 | 48±3 | -0.5±0.6 |
| | **Word Salad** | -0.5 | 31±1 | -0.1 | 43 | -0.3 | 40 | -0.2 | 44±1 | -0.2±0.3 |
| | APO | -0.4 | 51±1 | -1.0 | 57±1 | -0.6 | 51±1 | -0.1 | 48 | -0.5±0.7 |
| | CoT | -0.6 | 47±1 | -2.7 | 55±4 | -1.3 | 45±2 | -1.0 | 38±1 | -1.4±1.9 |
| | **SoT** | 0.1 | 20±1 | -0.6 | 25±1 | -0.1 | 26 | 0.5 | 22 ±1 | 0.0±0.6 |

Table 2: Slopes and accuracies averaged over tasks. The rightmost column has the average slope for all LLMs. Word salad and SoT are not factored into our main results, but are discussed in Section 6.1. The effectiveness of the prompts depended on the slope and $\sigma$: large $\sigma$ and a positive slope means an increasing trend in accuracy, with larger slopes implying a larger change. Shot slopes are positive, and the $\delta$ slopes are slightly negative. This suggests that more shots improve accuracy in all prompts; but in OOD this is ineffective, defaulting to the average and decreasing overall.

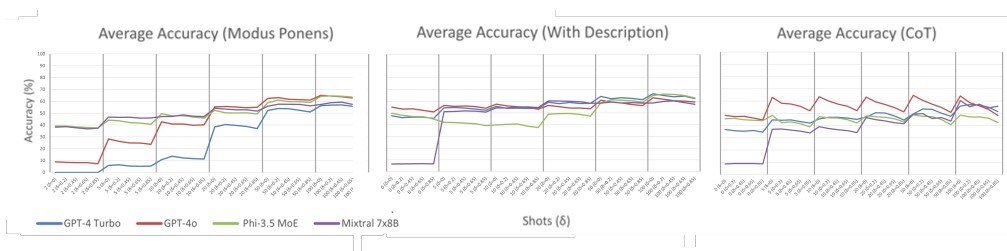

Figure 2: Per-model average accuracy results for (*left to right*) modus ponens, description, and CoT; plotted over shots (thick vertical lines) and per-shot $\delta$ between them. On average, most prompts showed analogous behaviour in the limit, and robustness to OOD. CoT also showed converging behaviour, although it was more brittle to OOD. Every datapoint is an average over 1,000 predictions.

## 5.2 DISTRIBUTIONAL SHIFTS

**Distributional shift decreased accuracy** as $\delta \to 0.85$. We evaluated the slope on the per-LLM accuracy averages between $\delta = 0.85$ and $\delta = 0$, per shot. All were negative. The largest (most sensitive to OOD) was CoT at -1.4, followed by APO (-0.5). The smallest was modus ponens, at -0.4 (Table 2). See Figure 2 for examples and Appendix G for a full breakdown. GPT-4o was most sensitive to OOD inputs, with an average slope of -1.2 (versus -0.7, -0.4, and -0.3 for Phi-3.5, Turbo, and Mixtral). The largest impacts of $\delta$ per task were in Reversal (-1.7±1.5), versus Vending Machine (Sum) (lowest; $0.1 \pm 0.2$). The average slope was -0.6±0.3.

## 5.3 FINE-GRAINED BEHAVIOUR

In the breakdown per-prompt and per-task, LLMs had (1) similar behaviours over the tasks, but (2) inconsistency over the task type. The first was given by the LLMs having low $\sigma$ but similar accuracy in a task-by-task and prompt-by-prompt basis: **all prompts had a positive slope and low relative difference among them** (Figure 4, in the Appendix). The per-prompt shot slopes, averaged per LLM, were 8.3±3.9 (modus ponens), 4.4±2.2 (description), 4.5±2.4 (DE), 5.3±2.6 (APO), and 3.3±2.4 (CoT) (Table 2). As per the averaged slope's $\sigma$, there was low variation between the type of LLM and the prompt over all tasks: 5.2±1.6 over shots. OLS fits over the per-shot $\sigma$ indicated that **the model gap, as the shots increased, narrowed**: -2.6±0.5; a similar pattern in the OLS fit

was noticeable in $\delta$ slopes. Inconsistency was when **related tasks had gaps in peak performances**: 31% (Maze (Solve) versus Pattern Matching), and 12% (Reversal and Stack; Table 1).

# 6 ABLATION STUDIES

We present summaries of our ablation studies. Refer to Appendix G for details and figures.

## 6.1 IMPACT OF LEXICAL FEATURES

We sought to understand to which extent lexicality (words) impacted ICL with respect to the *data features*. We assumed that LLMs were pretrained mostly on natural and programming languages. We compared word salad with modus ponens and DE; and CoT with SoT. While **word salad** versions of prompts started low–at some points with zero accuracy–they **quickly reached relatively high maximum accuracies**. Averaged per-LLM, **the word salad versions matched the baselines** to within $\sigma$ or $\sigma/2$ of their average and had the largest slopes. Word salad only randomised the system prompt, but SoT fully randomised the exemplars. It **had a major impact on accuracy**, with the lowest average performance over shots ($23\pm4\%$) in any prompt due to its high eror rate. **Some LLMs in SoT obtained above-average accuracies in certain tasks**, such as GPT-4o in PARITY (63% at 100 shots), and Turbo in Stack (76% at 50 shots).

## 6.2 POSITIONALITY OF EXEMPLARS

On every call, all exemplars so far were equiprobable and fixed throughout ('unshuffled'). Here, we randomised the position of the same exemplars within the prompt ('shuffled'), and also fully randomised the exemplars (drawn i.i.d. from the training set). There was a **small variation in accuracy when the same exemplars were shuffled versus unshuffled**, with the latter having slightly lower average accuracies and per-prompt slopes, albeit higher slopes per-LLM. The best-performing prompts for average accuracies when shuffled versus unshuffled were always the same. When fully randomising the exemplars, we only measured and compared GPT-4o. On average, **fully randomising the exemplars yielded lower accuracies**, and had lower shot and higher $\delta$ slopes.

## 6.3 IMPACT OF ALTERNATE DISTRIBUTIONS

We altered $\mathcal{P}$ in four setups: the fully randomised and shuffled exemplars from Section 6.2; an imbalanced distribution with *only* negative labels; and a corpus with uniformly at random labels (both test and train) as baseline. We only analysed and compared GPT-4o without Vending Machine (Sum). The **imbalanced scenario achieved higher average accuracies** than all setups, matching or outperforming the unshuffled baseline. However, in this case, the average $\sigma$ increased on every prompt and every setup. The random label baseline had better $\delta$ slopes than the unshuffled baseline. Of note is also CoT, which had negative shot slopes in all setups.

## 6.4 COMPLIANCE VERSUS LEARNING

We separated parsing errors ('compliance') from mislabelled instances ('learning') and re-calculated averages and slopes. Factoring out compliance increased perceived performance by understating or overstating magnitudes. For example, average shot and $\delta$ slopes were smoothed out, thus making– for example–CoT's sensitivity to OOD hard to spot.

# 7 DISCUSSION

As the 'training set' (i.e., the number of exemplars) grew, (1) LLM accuracy increased, and (2) the gap between LLMs and prompts narrowed. Both suggest that ICL as a learning paradigm **depends less on the LLM and prompt and more on** the ability to perform **autoregression**. However, accuracies were not consistent across similarly-related tasks: Pattern Matching (FSA) was effectively solved, while Reversal (PDA) and Maze (Solve; FSA) had low accuracies. This suggests that **autoregression is limited in its ability to solve tasks**. This could be related to the choice of prompt. However, we observed that while all prompts were sensitive to OOD, the best prompts (CoT and

APO) were both adaptive and more brittle. This suggests then that, although they are effective on leveraging the power of a PDA, they bias the learner towards the *observed* distribution. This, in turn, from the perspective of our framework, means that learning in ICL is not completely fulfilling the requirements from Equation 2. Thus, **autoregression's *ad hoc* encoding via the prompt is not a robust learning mechanism**. As an extreme example, recall that Vending Machine (Sum) had non-zero accuracy but near-zero slope regardless of number of shots, thus indicating complete inability to learn the task.

Indeed, in the limit, accuracies were similar regardless of language and exemplar distributions, provided that they remained fixed. Hence **ICL learns the *observed* $\mathcal{P}$, rather than fully generalising to the *unseeen* $\mathcal{Q}$**, since the fully randomised exemplars had lower accuracy than both the shuffled and unshuffled settings. Remark that the observed $\mathcal{P}$ did not change, and this phenomenon also could be explained as a manifestation of the bias-variance tradeoff. Given that the randomised labels baseline had lower $\delta$ slopes, **OOD brittleness is very dependent on ICL overfocusing on spurious features**. This is especially visible in CoT, which had consistently negative $\delta$ slopes across all variations of $\mathcal{P}$. While description-based prompts had the best *peak* accuracy, in the limit word salad reached equivalence with them. In SoT, some LLMs were still able to reach above-random scores in spite of the constant randomisation. This means that **autoregression can distinguish data features from lexical relations, but cannot fully identify *feature* relations within the data**. It also empirically confirms the remarks from Equation 4 that $p$'s contribution vanishes in the limit.

**Alternate explanations** could be (1) contamination, and (2) tokenisation. Contamination could explain the accuracy in Pattern Matching, perhaps due to the (easy) $\Sigma$, $\{a, b, c\}$. Other tasks, like Reversal, used more complex $\Sigma$ and had lower scores, so it could be argued that the LLMs had been pretrained in these tasks. However, good performances were also observable in Hamiltonian and PARITY; thus suggesting the ability to (almost) fully simulate an FSA, and not contamination. For (2), it could be said that an LLM trained on a task $A$ will not necessarily solve a similar $B$ if $\Sigma_A \neq \Sigma_B$ (cf., graph and maze traversals). It could also explain the results from Vending Machine (Sum): arithmetic skills are impacted by BPE (Singh & Strouse, 2024), the tokeniser which all LLMs studied implement. The implementation is not always the same. In the limit, LLM performance gap narrowed and thus tokenisation is not as relevant to ICL as the data features, although this only applied to decision problems, not arithmetic. Finally, our theoretical framework could impact our analysis of the conclusions. We argue that mathematically it is sufficient to define learning due to its ablation on the *nature* of the data and its focus on *learning as a process*. However, we refine and discuss this in Appendices C and D, including further explanations on the accuracy gap observed.

## 8   CONCLUSION

In this work we began by noting that, *mathematically*, ICL did constitute learning. However, we also noted that further work was required to characterise it beyond the standard assumptions and limitations of the literature. Our experiments thus accounted for prompting style and phrasing, natural language, number of shots, input and output distributions, contamination, and pretraining strategies. We found that, although *formally* **ICL is a form of learning, *empirically* it is relatively weak**. This is due our findings on its limitations and nuanced behaviours, different than originally reported. Concretely, in the limit, best-of average accuracies were given by 50-100 shots, and the differences amongst both LLMs and prompts decreased. Exemplar positionality, characterisation, labels, and wording were less relevant than the data features themselves–even in SoT, LLMs learned the task in spite of its constant randomisation. Nonetheless, ICL also overfocused on spurious features from the observed distribution. It also showed marked differences in supposedly-related tasks, and brittleness to OOD, especially in APO and CoT.

Our findings indicate that, for example, brittleness to OOD means that LLM performance will not be well-characterised by testing only a few prompts, as the performance observed may be spurious. Hence, research on LLM capabilities must be done with caution and transparency, testing multiple prompts, shots, and distributions. Future work should characterise reasoning models: we conjecture that they will do better in our setup; but will also have difficulties in complex tasks (e.g., context-sensitive languages), brittleness to OOD, and inconsistency over tasks. The latter also suggests that an open question remains on how to empirically measure what ICL does over what it can do; and then map it back to the theory while accounting for natural language factors not studied in this work.

## 9 ETHICS

Our work is a large-scale exploration of ICL over synthetic data. We are unaware of any potential misuse of this research, albeit we could have overlooked something. The volume of data in our work likely had a very high carbon footprint. While we argue that releasing the code publicly will have more benefits to the community than potential harms, we have included disclaimers discouraging running the full suite. We expect this work to be a *one-off* experimental work to determine ICL's feasibility as a learning paradigm, and thus our work focused on various open and closed models. This should make the work comprehensive enough to also discourage re-running the full suite. We discuss limitations of our work in Appendix A.

## 10 REPRODUCIBILITY STATEMENT

All code is included in the repository. It will be open-sourced under the MIT licence. Detailed methodology, included model versioning, is in Appendix F. Prompts are in Appendix H, and also in the repository. Work has been done in both closed-source and open-source models. We set the temperature to zero throughout to ensure further reproducibility.

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

## A  LIMITATIONS

One core limitation of our work is that LLMs are continuously updated, and this could make reproducibility difficult. To mitigate it, we have worked with open and closed-source LLMs, and provided detailed call parameters. Our evaluation is not cheap either: running synchronously a single task per LLM could and has taken months, depending on hardware, and the aggregate cost for all calls could render further exploration prohibitive. Lower-volume testing or fewer tasks could provide similar results, at the expense of statistical significance or ambiguity. Other testing, such as alternate paradigms (e.g., reasoning models) and prompting (multi-step, multi-call) were not evaluated in our work and could show more nuanced results.

Finally, interpreting the results from the ML baselines is nuanced. These are fast to train and iterate over, although they require larger amounts of data, and are also sensitive to OOD. We attribute this brittleness to the input representation length, which is characteristic of all tasks except PARITY. No neural networks beyond an MLP were tested. It is known that LSTMs and RNNs excel at these tasks (Butoi et al., 2025), albeit also require significant data volumes.

## B  DETAILED BACKGROUND

### B.1  PAC LEARNING

The original framework from Valiant (1984) centres itself on the learnability of the concept class, rather than the learner. We reproduce it here and compare it to our rephrasing to describe generalisation. For more thorough discussions on this framework, see Kearns & Vazirani (1994). For a formal, full description of in-context learning within the context of PAC learning, see Wies et al. (2023).

Suppose we wish to model a binary classification phenomenon with features assumed to be drawn from some nonempty set $X \subset \{0,1\}^m$ (the instance space). This phenomenon is labelled with a function (the concept) $c\colon X \to \{0,1\}$.[4] A set of concepts $C = \{c_1, \ldots, c_k\}$ is called a concept class. A learning algorithm is then tasked with classifying samples $x \sim \mathcal{P}$, where $\mathcal{P}$ is supported on $X$. It selects a hypothesis (another concept) $h$ such that

$$error(h) = \Pr_{x \sim \mathcal{P}}[c(x) \neq h(x)]. \tag{6}$$

To learn, the algorithm is given access to a function $\mathcal{O}\colon C \times X \to \{0,1\}$ that provides i.i.d. samples with some distribution $\mathcal{D}$. Then, a concept class $C$ over $X$ is PAC-learnable if an algorithm outputs an $h \in C$ such that

$$\Pr\left[error(h) \leq \epsilon\right] \geq 1 - \delta \tag{7}$$

for all $c \in C$, any $\mathcal{P}$, and $\epsilon \in (0, 1/2)$ and $\delta \in (0, 1/2)$, with an observed subset $D$ built with $\mathcal{O}$. The algorithm is required to run in $\text{poly}(m, |c|, 1/\epsilon, 1/\delta)$, Where the size $|c|$ is the smallest way to represent $c$ under a chosen map $R\colon \Sigma^* \to C$. Stochasticity is accounted in both calls to $\mathcal{O}$ and the learner's internal state.

Our framework has five core differences:

1. The learning algorithm is a machine learning *model*.

2. The selected hypothesis is the learner's weights as a function of the input, $f(x_k)$. Namely, accounting for autoregression, $f(z)$ for $z = f(x_{k-1})$.

3. The 'access' to $\mathcal{O}$ is replaced by a preconstructed dataset observed during the prompt call.

4. We reframe Equation 6 to work on the average *empirical* error of a dataset, to align it with contemporary evaluation methods.

5. We replace the concept class $C$ and the selection of said concepts with a single function $c$. This can be shown to be equivalent by composition by noting that $c$ can act as the selector for the concept class $C$.

---

[4]$X$ can also be an Euclidean space, $X \subset \mathbb{R}^m$. Concepts may also be equivalently seen as subsets of $X$.

The last difference renders our framework imprecise, but not weaker, when compared to standard PAC learning. PAC learning is known to have certain limitations. For example, deterministic finite automata and context-free grammars cannot be learnt in the standard PAC setting (Pitt & Warmuth, 1993; Kearns & Valiant, 1994; Niyogi, 2006); and contemporary neural networks has been shown to be able to learn beyond seen concept classes (see, e.g., Kawaguchi et al. 2022). Our reframing avoids these limitations by removing the dependence on a specific task (concept class) and instead assumes that a subset of $X$ is labelled with some $c$, in line with the expectations on current LLMs. More importantly, neither of the points above detract from the definition of learning as generalisation.

### B.2 AUTOMATA THEORY AND FORMAL LANGUAGE THEORY

A formal grammar $\mathcal{G}$ is a collection of strings from an alphabet $\Sigma = S \cup N \cup \{\epsilon\}$, and production rules (maps) between them. $S$ and $N$ are sets of non-terminal and terminal symbols, and $\epsilon$ the empty symbol, respectively. These rules form a set (language) $L$. Grammars may be categorised based on their complexity–namely, the production rules–using the Chomsky hierarchy. Each class is a proper subset of the other.

The classes of automata (functions) needed to answer whether $x \in L$ for a string $x$ have an isomorphism between them and the classes of formal grammars in the Chomsky hierarchy. More complex languages require more complex automata, whose classes are also supersets of the others. This is because every automaton, with the exception of the Turing machine, is limited in a certain way. For example, FSA read the input (tape) symbol-by-symbol in one direction and a single pass; change their internal state between accept and reject; and return either when the read is complete. They thus can recognise precisely the set of regular languages. Pushdown automata, or PDA, are equivalent to FSA but with a memory stack added, and can recognise context-free languages. See Appendix E for classifications of our tasks within the context of both automata and formal language theory. Remark that LLMs can perform recursion and feed their own 'pseudo-state' (namely, the token outputs) back into itself. This allows it to maintain a memory stack, albeit not fully controllable–hence why CoT is effective to a point: it templatises the memory stack.

## C  ALTERNATE MODELS OF LEARNING

In Section 3.1 we noted that PAC learning is the predominant model for learning in computational learning theory, and, specifically, learning theory. It is also frequently used when modelling language learning (see, e.g., Niyogi 2006). However, it is not the only model of learning, or the only accepted definition of learning. For example, Gold's inductive inference framework (Gold, 1967) is also sometimes used to model other language acquisition and learning (Johnson, 2004), and forms the basis of algorithmic learning theory.

Frameworks outside of computer science, such as those used in psychology and education, are also designed to formally define and measure learning. While the definition of learning has broad agreement (the ability to behave in a given way based on experience; Schunk 2012), measurement protocols differ amongst theories.

In this section we discuss alternate models of learning, from inside and outside of computer science. Specific algorithms and approaches, such as the Triggering Learning Algorithm (Gibson & Wexler, 1994) or back-propagation, are not covered here as they may be framed in a model of learning (e.g.,in terms of PAC learning Niyogi 2006). For non-computer science models, we focus on the two major theories with well-defined measurement protocols: behavioural and cognitive. We do not cover other frameworks, such as information theories of learning (Shuell, 1986), since they explicitly require the learner to encode and retrieve knowledge for arbitrary periods of time, and this is not the case for ICL.

### C.1  GOLD'S INDUCTIVE INFERENCE

In the inductive inference framework, the learner observes an infinite sequence of examples $x_1, \cdots \in \Sigma$ from some language $L$ generated by some grammar $\mathcal{G}$. The sequence may contain duplicate elements. Let $\mathcal{G}(x_k)$ be said sequence up to the $k^{\text{th}}$ element. It is said that a learner $f \colon \Sigma \to \Sigma$ *learns $L$* (and thus $\mathcal{G}$) *in the limit* if, based on a chosen metric $d \colon \Sigma \times \Sigma \to [0, 1]$,

$$\lim_{k \to \infty} d\big(\langle f(x_1), \ldots, f(x_k)\rangle, \mathcal{G}(x_k)\big) = 0. \tag{8}$$

Namely, it is said that $f$ has learnt $\mathcal{G}$ (in the limit; in the Gold sense), if after $k$ instances, the learner will correctly identify all observations from $L$. Remark that the choice of distance directly affects the definition, and that this learner *only* observes positive examples. In this framework, it follows thus that learning a language (r. concept in PAC learning) is equivalent to–eventually–perfectly reproducing the grammar. In contrast, in PAC learning, learning is probabilistic and can only be done w.p.1 in the limit.

It follows then that the main criticism to this framework is that it is too rigid, as it does not allow the learner to make mistakes. Reframings to allow for a looser distance or a threshold number of errors are effective at allowing more pragmatic learnability (Wharton, 1974). Other variants, such as query learning (Angluin, 1988) have also been shown to be equivalent to this framework (Lange & Zilles, 2005). Nonetheless, they are weaker than the original statement (Niyogi, 2006).

From a theoretical perspective, Gold (1967) showed that the languages represented by deterministic finite automata and context-free grammars are not learnable in the limit. As noted in Appendix B, PAC learning is also limited in its ability to learn certain formal problems. However, it is possible to create variants of one framework to learn languages that cannot be identified in the other (Niyogi, 2006). Thus, Gold's and Valiant's frameworks are distinct and non-equivalent.

In the context of our work, we are more concerned about measuring learning. Since Equation 6 is a distance metric, we may use our reframing equivalently in the Gold sense, and setting a (probabilistic) threshold $\epsilon$ as before,

$$\lim_{k \to \infty} \Pr\left[d\big(\langle f(x_1) \ldots f(x_k)\rangle, \mathcal{G}(x_k)\big) > \epsilon\right] = 0. \tag{9}$$

However, the statement around the probability of this threshold holding (Equation 7) would be missing, and hence the conclusions that we could draw are weaker.

## C.2 Behavioural Theories

Behavioural models of learning focus on how much feedback (reinforcement of correct guesses) as well as the developmental status of the learner. While the autoregressive mechanism could account for feedback, the developmental status is more difficult to approximate. We argue that this could be considered the pretraining process. We include a small experiment with an untrained model in Appendix D.

In the connectionist model of learning (Thorndike, 1913), learning is given by associations between experiences, and through trial and error. Experiments to measure this theory were carried across multiple months (e.g., participants had to close their eyes and draw a line of a specified length hundreds of times for several days). From our work's perspective, measurement was done within the same $\mathcal{P}$.

Another well-known model is that of operant conditioning Skinner (1938). This framework adapts closely to our work. Its full formulation includes the process by which learning occurs (e.g., conditioning, reinforcement, etc), and may be found in Schunk (2012). In this section we limit ourselves to describe the measurement itself. This is due through Skinner's definition of generalisation, which involves the repeated response to an input; and discrimination, which is varying the specific response based on the input. The core problem with generalisation in this theory is that, since learning relies on reinforcement, responses cannot be given *without* having been given previously said reinforcement (i.e., there cannot be zero-shot learning). The explanation for humans is that they rely on the composition of previously-learnt behaviours, and thus zero-shot learning may occur. For LLMs, this could *also* be argued based on the 'developmental status' of these learners: namely, the pretraining itself. Discrimination is also measured through zero-shot learning; namely, providing an appropriate response to an instance *after* being given a general description of the task.

Fitting LLMs into Skinner's framework means that generalisation is measured through repeated presentation of exemplars (and their correct labelling). Zero-shot *in this case* means observing only the instance of the problem and not having any feedback. Concretely, this was zero-shot modus ponens; which, as we observed, had near zero-performance across the board–as expected since **the tasks ablated for memorisation** and the learner had no reinforcement. On the other hand, discrimination requires the task description itself. This is more akin to zero-shot learning in the Description, CoT, and DE scenarios; and, to a minor extent, word salad and SoT.

What ties all these frameworks together to PAC learning is that theories have a certain tolerance to learner error, which in turn makes them closer to this framework than to Gold's.

## C.3 Cognitive Theories

In behavioural theories, learner variation is studied by evaluating the impact of the environment and the previous reinforcement steps. In contrast, cognitive theories emphasise how the differences between the prior knowledge of the learners, along with their own internal processes, impact learning. They also distinguish between learning and performance Schunk (2012). This means that, for example, in these frameworks, a learner could acquire latent knowledge by observing the environment although never actually obtaining reinforcement. From the perspective of our work, this is visible in all prompts minus modus ponens, *except* that only in the zero-shot setting. It is well-known that the way by which the learner is exposed to the task (e.g., demonstration, explanations, etc.), as well as the feedback (success) directly affects the effectiveness of the learning process (Schunk, 2012). Our work accounts for the first aspect (e.g., by the prompt style itself), but not the second. These, however, are more akin to how a reasoning model outputs text.

Ultimately, cognitive theories measure learning through success after reinforcement (or without, in the case of latent knowledge), as well as retention. Our work partially measure these. Same as in the behavioural theories, the tolerance to error makes these frameworks closer to PAC learning than to the inductive inference framework.

# D    RELATIONSHIP TO NATURAL LANGUAGE

## D.1    IMPACT ON CONCLUSIONS

We noted throughout our work that the synthetic nature of our work could underestimate the performance of an LLM on realistic scenarios. The choice of synthetic data was to ablate out the LLMs' intrinsic knowledge, and instead focus on their ability to infer features from the observed $\mathcal{P}$. It also allowed us to control every aspect of the data–from contamination to ID/OOD–to ensure a fully 'sanitised' experiment suite. This practise is known to be useful to study generalisation (Power et al., 2022).

However, learnings based on synthetic data do not necessarily fully translate to natural-language scenarios. This is because synthetic data setups overlook considerations ubiquitous to natural language, such as compositionality, feature distribution, and ambiguity. These are all encoded–in one way or another–in pretrained LLMs.

Even when dealing with natural-language problems which are fully unseen by an LLM (for example, a language isolate), computational complexity comes into play. It is known that, under certain assumptions, natural language lies somewhere between context-free and context-sensitive grammars (Jäger & Rogers, 2012), which, in turn–as per our results and the theoretical work from Section 3– makes these problems difficult to solve without any prior knowledge. On the other hand, the vast literature and success stories of LLMs suggest that further empirical work is required to characterise what these models *do*, not what they *could do*.

Thus, our results are limited to the ability of ICL to draw conclusions from the data's features *alone*, eschewing any potential semantic priors induced by natural language. They must be interpreted with caution when considering their extension to natural language, particularly in tasks and evaluations which could rely on a model's latent knowledge.

## D.2    IMPACT ON RESULTS

To follow the point on latent knowledge, we remark that a full evaluation of ICL should account for an inductive bias-free learning. This means that the models must not have seen *any* of the data before, including natural language.[5] In line with the empirical spirit of this work, and in order to confirm this, we ran the same experiments for PARITY, Pattern Matching, both Vending Machines, and Hamiltonian in all shots and $\delta$. The learner was a separate, *randomly-initialised* model (Qwen 2 1.B Instruct; Yang et al. 2024). The model had accuracy zero in every task and setup, consistently showing responses such as 'itian常常uzzle' and '披露との披露', and thus having 100% error rate regardless of shots. A brief examination of the finetuned model revealed consistent, albeit not necessarily accurate, responses (e.g., 61.6%, 73.5%, and 52.9% for modus ponens ID at 20 shots in PARITY, Pattern Matching, and Reversal, respectively).

The above aligns with results from the literature and our work, but also opens further areas of research. Namely, it is known that priors are needed for ICL (Chang & Bisk, 2025; Hu et al., 2025). Also, within our setup, we found that an LLM's linguistic capabilities do not impact ICL (ref. word salad and SoT), and that a sufficiently large number of exemplars suffices. A gradual comparison of the learning (pretraining) process of an LLM with respect to its ability to understand data features would provide much needed information as to which extent the natural-language data impact ICL as a learning mechanism.

---

[5]Recall from Appendix C that, from the perspective of some theories, full zero-shot is not possible.

# E  FULL TASK DESCRIPTIONS

In this section we describe each task more precisely, and are summarised in Table 3. For concrete code examples, see the repository and Appendix H.

**PARITY**  Decide if a given binary string $\{0,1\}^k$ has an even number of zeros. Here, $\Sigma = \{0,1\}$ and the automaton decides whether to append $x \in \Sigma$ based on the transition probabilities. Emission is given by a fixed probability of $\frac{1}{10}$. Unlike most problems, PARITY's average length per $\delta$ was relatively fixed, at 19 characters. The difference was the probability of each character occuring in sequence. PARITY is classified as a regular language and modellable with an FSA.

**Pattern Matching**  Decide if a pattern $abcabb$ is a substring of a given string $x \subset \Sigma^*$, where $\Sigma = \{a, b, c\}$. The automaton is similar to PARITY's, with transition probabilities fixed by state ($x \in \Sigma$) but dependent on $\delta$. Strings with less than eight characters where rejected. In OOD scenarios, the sequence length grew to over five times the ID length. Pattern Matching is classified as a regular language and modellable with an FSA.

**Reversal**  Given a string of the form $l\#r$, the goal is to decide if $l$ equals the reversed $r$, $l = r^{-1}$. The start of $r$ is given by the delimiter $\#$. Same as PARITY, the selection of every string depends on transition probabilities $\delta$. In this case, the alphabet was picked to *not* be grammatical, $\Sigma = \{$gfx, chtte, %, ltintprk, ¯\\_(ツ)_/¯, start$\} \cup \{\#\}$ where $l, r \subset \Sigma^k \backslash \{\#\}$. In OOD scenarios, the sequence length grows to over seven times the ID length as $\delta$ increases. This variant of Reversal is a DCF language modellable with a PDA (Butoi et al., 2025).

**Stack**  For a final string $s_f$, starting string $s_0$, and series of operations $Op$ on a string, decide if $s_f = Op(s_0)$. The operations simulate a stack (push/stop/pop) and may or may not be grammatical (e.g., stack overflows). Same as PARITY, the selection of every string depends on transition probabilities $\delta$. Here $\Sigma = \{0,1\} \cup \{$push, pop, stop, empty$\}$, $s_f, s_0 \subset \{0,1\}^k$, and $Op \subset \Sigma^k \backslash \{0,1\}^k$. In OOD scenarios, the sequence length grew to almost three times the ID length as $\delta$ increased. Stack is a DCF language modellable with a PDA (Delétang et al., 2023).

**Hamiltonian**  Given a directed graph in adjacency matrix form $G$, and a path $p$, decide if $p$ is Hamiltonian. Under this setup, this problem is classified as a regular language and modellable with an FSA (Barrett et al., 2000). In OOD, the edges, and not the vertices, grew to up to 20% the original length. Consequently, the character description of the graph grew by up to 32%, from 695 characters to 851.

**Maze (Complete and Solve)**  Given a maze, two segments of the solution path, and a sequence of moves, in Maze Complete the task is to determine if the moves connect both segments. The separation between segments–but not the move sequence–is never longer than three moves. Maze Solve is given the full path and a longer sequence of moves. The task is to determine whether these moves lead to the solved maze (a valid path from start to exit). Both problems are classified as regular languages and modellable with FSA (Barrett et al., 2000). In OOD scenarios, the maze size became larger, albeit the average path length remained somewhat stable.

**Vending Machine (Verification and Sum)**  Given a list of items and costs $C$, a sequence of operations $Op$ (add balance, purchase item), and initial and final balances $b_0, b_f$, verify if $b_f = Op(C) + b_0$ (verification) or compute $b_f + Op(C) + b_0$ (sum). For the purposes of this problem, the items and costs were given in natural language: biscuits cost 20, soda costs 25, and coffee costs 15. Here $\Sigma = \{+20, +15, +25\} \cup \{$coffee, biscuit, soda$\}$, or, without resorting to strings, the abelian group $A_{vm} = (\{0, 20, 15, 25\}, +)$. The first three states denote additions, the named states are subtractions (item purchases), and the last state is the final balance $b_f$. Same as PARITY, the selection of every string $s \subset \Sigma^k$ depends on the transition probabilities $\delta$. In Vending Machine (Verification), the learner must assert if the last part of the string, $b_f$, equals the sequence of operations. Hence, it is a regular language and modellable with an FSA. Since the strings are always up to length $n$, $A_{vm}^n$ is a finitely-generated abelian group, and thus the decision version of Vending Machine (Sum) is a DCF language modellable with a PDA: it can be reduced to Stack with homomorphisms between the operations (e.g., push and pop versus add and subtract, respectively), and between the inputs

| Task | Label Balance | Average Lengths | Class |
|---|---|---|---|
| PARITY | 49, 50, 50, 50, 50 | 18, 17, 17, 17, 17 | FSA |
| Pattern Matching | 50, 50, 50, 49, 50 | 40, 46, 62, 92, 179 | FSA |
| Reversal | 49, 50, 50, 50, 50 | 86, 186, 220, 312, 567 | PDA |
| Stack | 50, 50, 50, 49, 50 | 97, 169, 207, 235, 263 | PDA |
| Hamiltonian | 50, 50, 50, 50, 50 | Graphs: 695, 862, 773, 770, 851 | FSA |
| | | Vertices: 10, 12, 11, 11, 12 | |
| | | Paths: 24, 27, 26, 26, 28 | |
| Maze Complete | 50, 50, 50, 50, 50 | 174, 173, 173, 175, 178 | FSA |
| Maze Solve | 50, 50, 50, 50, 50 | 429, 414, 423, 459, 498 | FSA |
| Vending Machine (Verification) | 50, 49, 49, 49, 49 | 105, 104, 111, 118, 128 | FSA |
| Vending Machine (Sum) | – | – | – |

Table 3: Label balances (as an average of positive entries) and description (string) lengths for values of $\delta \in \{0, 0.2, 0.45, 0.65, 0.85\}$. Every length depends strongly on the design of the automaton: some lengths grow much more slowly than others (e.g., PARITY versus Reversal). Other depend on the complexity of the task, as opposed to the input description length. For example, Hamiltonian maintains a relatively stable average number of vertices, but the connectedness of each graph increases with $\delta$. Vending Machine (Sum) is not classed here because it is not a decision problem.

($A_{vm}^n$'s set and, say, $\{00, 01, 10, 11\}$). In practice, the number of possible outputs is finite (albeit very large), but it requires the learner to keep track of a state. In both OOD scenarios, the sequence length became longer, by up to 20%.

# F    DETAILED METHODOLOGY

## F.1    LLM CALL PARAMETERS

We tested four LLMs: GPT-4 Turbo, GPT-4o, Mixtral 8x7B instruct v01, and Phi-3.5 MoE Instruct. Details for each model are in Table 4.

| Model | Description |
|---|---|
| GPT-4 Turbo$^\times$ | OpenAI model with a context window of 128k tokens. Version: GPT-4-0125. |
| GPT-4o$^\times$ | OpenAI model, higher-performing when compared to GPT-4 Turbo, and with a 128k context window. Version: GPT-4-0125 |
| Phi-3.5-MoE-Instruct | Mixture-of-experts model with a 128k context window and 6.6B active parameters. |
| Mixtral-8x7B instruct v01 | Mixture-of-experts model with a 32k context window and 12.9B active parameters. |

Table 4: Models evaluated. For the models marked with $\times$, details regarding architecture, parameter size, or pretraining strategies have not been disclosed. All models are instruction-pretrained.

All models were called with temperature set to zero and maximum return tokens of 3 for all prompting strategies, except CoT (1,024) and the system prompts generated by APO (512).

The APO algorithm was called with a batch size of 1024, beam width 4, and a search depth of 6.

All work was done on a Standard_ND40rs_v2 instance in Azure, which is equipped with eight NVIDIA Tesla V100 GPUs with 32 Gb of memory each. Calls were made using either the Azure Open AI API (OpenAI models only) or calling directly the models on the instances. Every model was called up to five times to account for any potential parsing errors or rate limitations from APIs. The data analysis was carried out on a consumer-grade laptop.

## F.2    BASELINES

The baselines were implemented in scikit learn (Pedregosa et al., 2011). For all tasks, the parameters were left as default and used as a random seed 13213. For every entry the string-based representation was mapped to integers character-by-character. That is, for, example, 'ltintprk' from Reversal was mapped to 4. Operators (e.g., '+' or 'pop') also mapped to integers. Since most models required tapes of the same length, empty cells were mapped to $-100$. No simulations of state (e.g., the state of the stack after a push) were included in the tape.

# G DETAILED RESULTS

## G.1 MAIN RESULTS

The full results of our main results are in Figure 3. It can be observed from the prompts that LLMs generally present the same average per-task behaviour, with minor changes depending on the prompt. In Table 5 we present the results per LLM and task excluding Vending Machine (Sum).

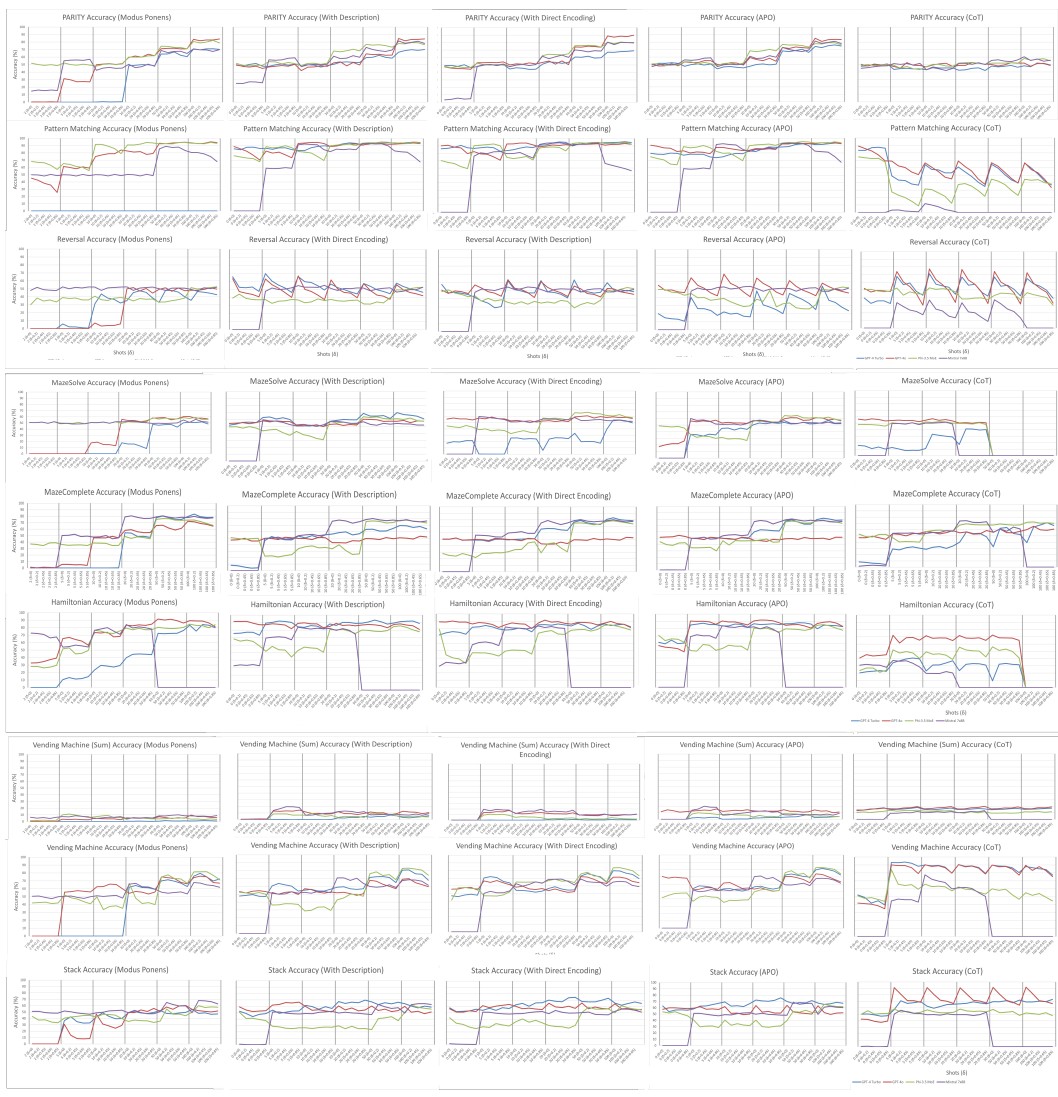

Figure 3: Complete set of performances per problem, including averages at the top. Observe how the averages do not necessarily correspond to the performance per-model per-prompt per-task. Consistent behaviours are that CoT is not robust to OOD, and that tasks on average present the same approximate behaviour regardless of prompt.

| | Prompt | Turbo | | GPT-4o | | Phi-3.5 | | Mixtral | | Avg. slope for acc. |
|---|---|---|---|---|---|---|---|---|---|---|
| | | Slope | Acc. | | | | | | | |
| **Shots** | Modus Ponens | 14.6 | 31±25 | 12.7 | 47±22 | 5.9 | 56±10 | 5.7 | 53±10 | 9.1±2.9 |
| | Description | 3.3 | 61±5 | 1.5 | 63±3 | 6.2 | 55±11 | 9.8 | 51±21 | 5.0±1.9 |
| | DE | 11.6 | 34±21 | 13.5 | 47±24 | 12.9 | 43±23 | 11.7 | 47±22 | 5.2±1.9 |
| | **Word Salad** | 6.7 | 57±12 | 2.3 | 62±4 | 5.1 | 57±10 | 10.3 | 51±21 | 12.3±3.1 |
| | APO | 4.2 | 50±7 | 1.8 | 60±4 | 0.8 | 49±1 | 8.2 | 39±15 | 6.1±2.0 |
| | CoT | 2.1 | 21±4 | 3.6 | 26±7 | -0.1 | 27±4 | 2.0 | 22±5 | 3.6±2.3 |
| $\delta$ | Modus Ponens | -0.3 | 31 | -0.6 | 47 | -0.6 | 56 | -0.3 | 53 | -0.4±0.4 |
| | Description | -0.5 | 61 | -1.1 | 63±1 | -0.4 | 55 | -0.2 | 51 | -0.6±0.4 |
| | DE | -0.6 | 34 | -0.1 | 47 | -0.3 | 43 | -0.3 | 47 | -0.5±0.6 |
| | **Word Salad** | -0.5 | 57 | -1.1 | 62±1 | -0.6 | 57±1 | -0.1 | 51 | -0.3±0.3 |
| | APO | -0.8 | 50±1 | -3.1 | 60±4 | -1.5 | 49±2 | -1.0 | 39±1 | -0.6±0.7 |
| | CoT | 0.0 | 21±1 | -0.7 | 26±1 | -0.2 | 27 | 0.3 | 22 | -1.5±1.9 |

Table 5: Slopes and accuracies for every LLM, averaged over prompts and tasks, excluding Vending Machine (Sum). On the rightmost column is the average slope for all LLMs. Rows in bold (word salad and SoT) are not factored in our main results, but discussed in Section 6.1. The numbers changed when compared to Table 2, but not substantially, thus leaving our results unchanged.

## G.2 Fine-Grained Behaviour

As mentioned in the main section, when breaking down the results per-prompt and per-task, the LLMs had (1) similar behaviours over the tasks, but (2) inconsistency over the task type.

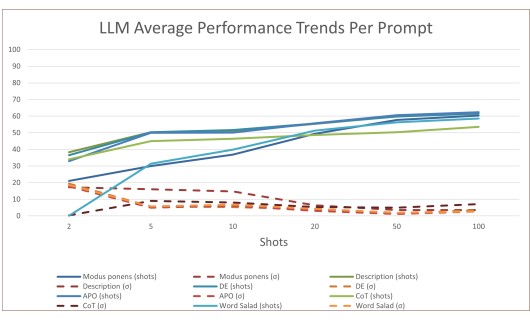

Figure 4: Averaged over all tasks and models, all prompts have a positive slope (5.2±1.6) over shots, and a narrowing gap in their $\sigma$ (-2.6±0.5).

Behavioural similarity was given by the LLMs having low $\sigma$ but similar accuracy in a task-by-task and prompt-by-prompt basis: **all prompts had a positive slope and low relative difference among them** (Figure 4, left). Indeed, the per-prompt shot slopes, averaged per LLM, were 8.3±3.9 (modus ponens), 4.4±2.2 (description), 4.5±2.4 (DE), 5.3±2.6 (APO), and 3.3±2.4 (CoT) (Table 2). The average of the slopes over shots is 5.2±1.6. We can hence observe that there was low variation ($\sigma$) between the type of LLM and the prompt over all tasks, and that the overall trend for all models, tasks, and shots is positive. An OLS fit over the per-shot $\sigma$ indicated that **the model gap, as the shots increased, narrowed**: -2.6±0.5. This is visible in the image to the left, where the models start with a large gap on every task (namely, word salad and modus ponens), which narrows as shots increase. It is worth noting how CoT has a slight improvement, as noted earlier in its slope, but it remains relatively even when compared to higher-performing prompts (e.g., APO, DE). Indeed, the dotted lines in the image to the left, denoting the $\sigma$ over the slopes, indicates that most prompts progressively narrowed their differences in performance, although, again, CoT remained relatively steady. At a minor extent, this gap on average also narrowed in aggregates over $\delta$: -0.2±0.2.

Inconsistency over the task was visible after observing that **related tasks had gaps in peak performances**: 31% (Pattern Matching versus Maze (Solve)), and 12% (Reversal and Stack; Table 1). This is particularly important given that an all-purpose solver (e.g., a universal FSA) should be able to, theoretically, have perfect performance on all tasks of the same class (r. regular languages). While it is a stretch to expect that from an LLM, it is worth pointing out that their general-purpose generalisability is hence (naturally) limited. It then follows that ICL as a learning process depends strongly on the features observed in-distribution. We cover this further in Section 7. The remaining ablation studies focused on evaluating this hypothesis.

### G.3 ABLATION: IMPACT OF LEXICAL FEATURES

Word salad prompts started with low, and sometimes zero, accuracies. In the limit, however, all prompts matched the average best-of non-salad performances to up a $\sigma$, with the exception of Reversal and Vending Machine (Sum). In the case of PARITY, Pattern Matching, Maze (Complete), and Vending Machine (Verification), the match was within $\sigma/2$ (Table 6). This improvement was fast, with slopes of 9.8, 12.1, 11.6, and 9.8 (Turbo, GPT-4o, Phi-3.5, and Mixtral, respectively), for an average of 11±4.6. Compare with the slopes for description (4.4±2.2), DE (4.5±2.4), and modus ponens (8.3±3.9). On average, word salad prompts were the most robust to $\delta$, with values of -0.2±0.3 (versus -0.5±0.4, -0.5±0.6, and -0.4±0.4, respectively), albeit all were within the baseline $\sigma$. See Figure 5 for a side-by-side depiction of the prompts with respect to their non-salad equivalents (description and CoT).

| Problem | Highest | | Lowest | | Highest (Word Salad) | Shots |
|---|---|---|---|---|---|---|
| PARITY | 80±3 | 100-APO | 16±20 | 2-m.p. | 80± 5 | 100 |
| Pattern Matching | 94±1 | 50-DE | 24±20 | 5-CoT | 92±3 | 50 |
| Reversal | 61±11* | 100-CoT | 20±21 | 2-m.p. | 51±1 | 100 |
| Stack | 73±14* | 50-CoT | 20±21 | 2-m.p. | 56±13 | 100 |
| Vending Machine (Ver.) | 81±12 | 10-CoT | 22±22 | 2-m.p. | 78 ± 6 | 100 |
| Maze (Complete) | 77±5 | 100-m.p. | 9±16 | 2-m.p. | 74±6 | 100 |
| Maze (Solve) | 63±5 | 50-desc. | 17±20 | 0-APO | 54± 6 | 50 |
| Hamiltonian | 89±2* | 100-desc | 29±8 | 0-CoT | 68 ± 20 | 20 |
| Vending Machine (Sum) | 16±1 | 5-CoT | 0 | 0-DE† | 8 ± 2 | 100 |

Table 6: Highest and lowest accuracies, averaged by model. An asterisk denotes an average over fewer models (always excluding Mixtral); and † means that there were multiple ties. Highlighted in grey are the prompts where word salad match within a $\sigma$ the average best-of accuracy from the non-word salad prompts, and in blue these within $\sigma/2$. In most cases, the match occurred at 100 and 50-shot, except in Hamiltonian, where the highest best-of was attained at 20 shot.

Unlike word salad, SoT had a major impact on accuracy, and had the lowest average performance over shots (23±4) in any prompt. This was due to SoT's high parse error rate over almost all shots. In contrast, description had near-zero error rates, and modus ponens and word salad quickly converged to zero. Overall average shot and $\delta$ slopes in SoT hovered around zero (1.6±2.2 and 0.0±0.6, respectively). This does not imply the LLMs were unable to solve all problems under SoT. Some LLMs in SoT obtained above-average peak accuracies in certain tasks: GPT-4o in PARITY (63% at 100 shots), and Turbo in Stack (76% at 50 shots). However, high-performing problems like Hamiltonian and Pattern Matching had 14±12% and 2±3% average accuracies, respectively.

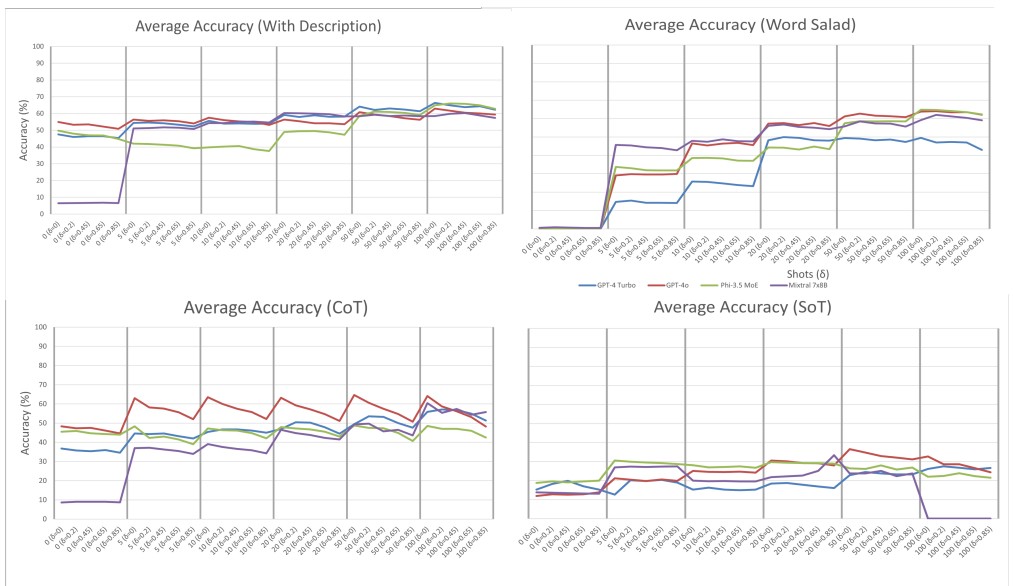

Figure 5: Average over all LLMs and tasks for the non-salad (left) and salad (right) prompts. Description-based prompts rarely performed poorly at zero-shot for all LLMs but Mixtral, while word salad versions required five shots (Mixtral), ten (GPT-4o), or more. However, word salad prompts eventually reached equivalence with their baselines (Table 6). In high-accuracy tasks (Hamiltonian, Maze (Complete) and PARITY) the prompts matched DE and modus ponens at between 10 and 100 exemplars. On the other hand, CoT and SoT had different behaviours: CoT had an (average) modestly increasing trend which was not reproduced in SoT. However, this is an aggregate: tasks such as Reversal had the same brittleness to OOD than their CoT counterparts; and tasks such as PARITY even showed above-random best-of accuracies in some tasks (Table 2).

## G.4 ABLATION: POSITIONALITY OF EXEMPLARS

In all our experiments, all exemplars were equiprobable and fixed throughout the experiment (*unshuffled*). In this experiment we randomised the position of the same exemplars within the prompt (all prompts and LLMs; *shuffled*), and *fully randomised* the exemplars per call by drawing them i.i.d. from the training set.

We observed a small variation on accuracy when the same exemplars were shuffled versus unshuffled (Table 7). The latter had lower performances, and larger slopes per-LLM, these were 5.6 versus 5.8 (Turbo), 5.2 versus 7.3 (Mixtral), 4.2 versus 4.2 (Phi-3.5), and 3.1 versus 3.5 (GPT-4o) (shuffled and unshuffled slopes). However, *per-prompt*, these slopes were higher. The best average accuracies in the shuffled setting were always with the same best-performing prompt from the unshuffled case, and within the reported $\sigma$ (e.g., 64±12% shuffled versus 61±11% unshuffled for Reversal at 100-CoT).

When fully randomising the examples, we only measured and compared GPT-4o (Table 8). Similar to the previous experiment, we observed variations on average and highest accuracies (e.g., 94%, 92%, and 93% highest for fully random, shuffled, and unshuffled, respectively, in Hamiltonian with description) although inconsistent (91%, 92%, and 71% in Stack CoT; 77%, 90%, and 90% in PARITY APO and DE). On average, however, fully randomising the labels yielded lower average accuracy (43%) versus shuffled and unshuffled (48% for both), and lower per-prompt accuracy.

| | Prompt | Turbo | | GPT-4o | | Phi-3.5 | | Mixtral | | Avg. slope |
|---|---|---|---|---|---|---|---|---|---|---|
| **Shots** | Modus Ponens | 12.8 | 34±22 | 10.2 | 44±18 | 5.6 | 50±10 | 3.9 | 50±7 | 8.9±4.0 |
| | Description | 3.6 | 57±6 | 1.4 | 56±3 | 4.6 | 49±10 | 6.3 | 51±14 | 4.7 ± 2.1 |
| | DE | 3.6 | 55±6 | 1.0 | 59±2 | 5.8 | 49±10 | 5.8 | 50±17 | 4.9±2.5 |
| | **Word Salad** | 8.8 | 28±16 | 12.1 | 43±22 | 11.4 | 41±21 | 9.1 | 45±19 | 11±5.0 |
| | APO | 4.3 | 54±8 | 2.0 | 57±4 | 4.5 | 50±9 | 2.1 | 56±4 | 5.4± 2.6 |
| | CoT | 3.7 | 49±7 | 1.3 | 56±4 | 0.6 | 45±1 | 7.0 | 39±13 | 3.6±2.6 |
| | **SoT** | 1.5 | 20±4 | 2.9 | 25±6 | 0.3 | 26±4 | 0.0 | 26±5 | 0.5±2.2 |
| **OOD** | Modus Ponens | -0.9 | 34±1 | -0.5 | 44±1 | -0.4 | 50±1 | -0.2 | 50±1 | -0.4 ± 0.3 |
| | Description | -0.3 | 57±1 | -0.8 | 56±1 | -0.6 | 49±1 | -0.1 | 51 | -0.5 ± 0.6 |
| | DE | -0.4 | 55±1 | -1.0 | 59±2 | -0.4 | 49±1 | -0.1 | 50 | -0.5±0.7 |
| | **Word Salad** | -0.5 | 28±1 | -0.2 | 43 | 0.0 | 41 | -0.3 | 45±1 | -0.3±0.3 |
| | APO | -0.2 | 54 | -1.0 | 57±1 | -0.6 | 50±1 | 0.0 | 56 | -0.6±0.6 |
| | CoT | -1.1 | 49±2 | -2.7 | 56±4 | -1.1 | 45±2 | -1.0 | 39±1 | -1.4±1.8 |
| | **SoT** | -0.2 | 20±1 | -0.6 | 25±1 | 0.0 | 26 | 0.0 | 26 ±1 | -0.2±0.6 |

Table 7: Slopes and average accuracies for shots and $\delta$, per prompt, with shuffled exemplars. Greyed out are the accuracies where the slope or accuracy was higher than the non-shuffled version from Table 2, but the $\sigma$ was higher. In blue are the setups with higher accuracy *and* lower $\sigma$. *Top to bottom*: average accuracies per-prompt were 44±14%, 53±7%, 53±8%, 39±19%, 54±6%, 47±6%, and 24±3%.

### G.5 ABLATION: ALTERNATE DISTRIBUTIONS

Due to cost, this section is limited to GPT-4o. In this experiment we altered $\mathcal{P}$ to show different distributions to the model. These alterations were:

1. Randomised exemplars on every call. This is the same setup from Appendix G.4.
2. Fully random exemplars drawn i.i.d. from the training set on every call. This is also the setup from Appendix G.4.
3. An imbalanced distribution of labels, showing *only* positive labels as exemplars.
4. A corpus with uniformly at random labels (both test and train) acting as a baseline.

The results are in Table 8. With the exception of the random baseline, **all setups showed the ability to learn the underlying distributions**: shuffled exemplars and the imbalanced scenario often matched or outperformed the baseline average accuracy over most tasks and prompts, with the latter attaining higher average accuracies and larger $\delta$ and shot slopes. However, the $\sigma$ in the per-prompt accuracies were higher. The baseline had an average accuracy of 41±9%: most prompts stayed within the random-choice 50±5% accuracy, with the exception of SoT (24%). Remark that this is expected, since every datapoint has the same probability of having any labels, and this is uncorrelated to the features themselves–i.e., it is unlearnable.

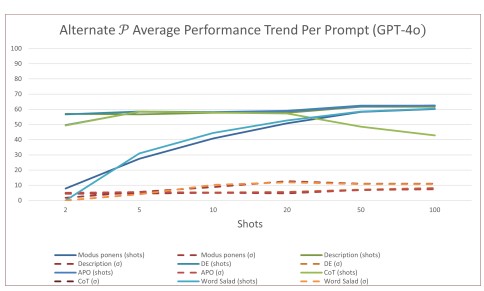

Figure 6: Average accuracies for alternate $\mathcal{P}$. Most prompts had positive slopes, with the exception of CoT, which peaked fast and then started decreasing.

As in all previous experiments, in the limit we observed increasing trends in the shot-slopes and $\delta$-slopes across setups and prompts, with average slopes of 4.3±4.2 and -0.8±0.6 (Figure 6, left). These were more noticeable in the shuffled setting, where all prompts but DE had improvements. However, in this case, the average $\sigma$ increased for both slopes: 1.0±0.7 and -0.3±0.2, respectively. This was observed on every prompt and every setup, as most setups learnt the problem. Of note is also CoT, which showed decreasing shot slopes across all setups: -1.1 for imbalanced labels, -1.1 for shuffled, and -2.5 for both fully random exemplars and random labels.

| | Prompt | Imb. | labels | Fully exemp. | rand. | Rand. | labels | Shuffled | | Avg. slope |
|---|---|---|---|---|---|---|---|---|---|---|
| **Shots** | Modus Ponens | 12.6 | 48±22 | 8.6 | 29±14 | 7.7 | 38±14 | 12.7 | 47±22 | 8.3±3.9 |
| | Description | 1.9 | 63±3 | 1.1 | 59±3 | 0.1 | 49 | 1.4 | 62±2 | 4.4±2.2 |
| | DE | 2.1 | 64±3 | 0.9 | 60±2 | 0.2 | 49 | 1.5 | 63±3 | 4.5±2.4 |
| | **Word Salad** | 13.4 | 49±25 | 9.1 | 30±16 | 8.9 | 37±17 | 13.5 | 47±24 | 11±4.6 |
| | APO | 2.3 | 62±4 | 3.3 | 58±6 | 0.8 | 49±2 | 2.3 | 62±4 | 5.4±2.6 |
| | CoT | -1.1 | 56±6 | -2.5 | 54±7 | -2.5 | 41±4 | -1.1 | 56±6 | 3.3±2.4 |
| | **SoT** | 2.9 | 26±6 | 2.2 | 25±5 | 2.2 | 24±5 | 5.1 | 24±7 | 1.6±2.2 |
| $\delta$ | Modus Ponens | -0.5 | 48 | -0.6 | 29 | -0.4 | 38 | -0.6 | 47 | -0.4±0.4 |
| | Description | -1.0 | 63±1 | -0.7 | 59 | -0.2 | 49 | -0.9 | 62±1 | -0.5±0.5 |
| | DE | -1.1 | 64±1 | -0.7 | 60±1 | -0.2 | 49 | -1.1 | 63±1 | -0.5±0.6 |
| | **Word Salad** | -0.0 | 49 | -0.3 | 30 | -0.0 | 37 | -0.1 | 47 | -0.2±0.3 |
| | APO | -1.2 | 62±1 | -0.7 | 58±1 | -0.2 | 49 | -1.1 | 62±1 | -0.5±0.7 |
| | CoT | -3.0 | 56±4 | -2.6 | 54±3 | -1.3 | 41±1 | -2.9 | 56±4 | -1.4±1.9 |
| | **SoT** | -0.7 | 26±1 | -0.7 | 25 | -0.7 | 24±1 | -0.4 | 24 | 0.0±0.6 |

Table 8: Slopes and average accuracies for shots and $\delta$, per prompt, on our evaluation of alterations of $\mathcal{P}$. Greyed out are the accuracies where the slope or accuracy was higher than *or equal to* the non-shuffled version of GPT-4o's predictions; and in blue these where the $\sigma$ was also strictly larger than GPT-4o's equivalent (Table 2; but excluding Vending Machine (Sum)). *Top to bottom*: average accuracies per-prompt were 37±17%, 53±2%, 54±2%, 37±19%, 53±4%, 52±3%, and 25±6%.

### G.6    ABLATION: COMPLIANCE VERSUS LEARNING

| Prompt | MP | Desc | DE | APO | CoT |
|---|---|---|---|---|---|
| Compliance | 43±10 | 53±7 | 53±6 | 52±6 | 46±8 |
| Learning | 53±6 | 56±3 | 58±4 | 57±3 | 56±6 |

Table 9: Average accuracy across all tasks and LLMs aggregated by prompt: it is slightly above average when not accounting for parsing errors. Large drops occur otherwise, with increases (up to double) in $\sigma$. This suggests that LLM comparisons depend strongly on measurement.

The distinction between compliance with the prompt (returning a parseable output) versus learning the task (returning a correct label) requires further scrutiny.

This is because, in the extreme case, a dataset could be, for example, 99.9% parsing errors and one lucky guess, thus leading to inaccurate assessments of performance. Hence, we separated parsing errors from mislabelled instances, and re-calculated the averages and slopes. **Factoring out parsing errors increased the perceived performance of an LLM**, usually understating or overstating magnitudes. This is because it becomes harder for an LLM to have accuracies below 45% (Table 9, left), and average shot and $\delta$ slopes are smoothed out (4.7±3.1 and -0.4±0.2 average, respectively) when compared to compliance and learning (r. 5.4±3.1, -0.5±0.3), thus making–for example–CoT's sensitivity to OOD hard to spot (Figure 7). In turn, this suggests that works should disclose the parsing strategies to avoid misleading results.

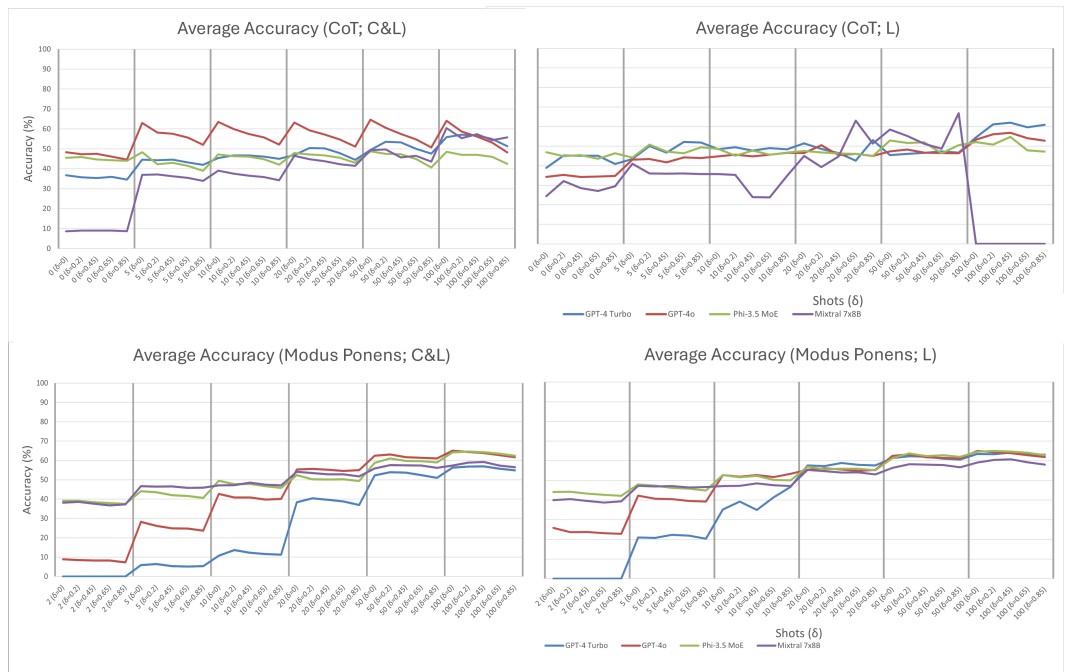

Figure 7: Comparison of select prompts when measuring reasoning as compliance *and* learning (*left*, labelled as C&L and counting parsing errors as mislabels), and only learning (*right*; not factoring in parsing errors) for CoT (*top*) and modus ponens (*bottom*). In learning, CoT's sensitivity to OOD is hard to spot as the performance curves are smoothed out. In comparison, when evaluating compliance and learning, the evaluation has a smoother performance floor and accounts for the full dataset. Model convergence was still noticeable, as evidenced by the modus ponens plots.

# H   PROMPTS

We show sample prompts for various problems and prompting strategies. For the full prompts, refer to the repository. In Prompt 1 we show the modus ponens and description prompts for PARITY; and in Prompt 2 a sample CoT for Pattern Matching. We also show in Prompts 3 and 4 the CoT and SoT versions of Maze Complete, and in Prompt 5 the word salad version of Vending Machine (Solve).

---

This task is called PARITY. The strings in PARITY are generated from a probabilistic automaton. Your job is to learn what is the likelihood of a string to be labeled 0 or 1, and output the correct label.

In the limit where the automaton is deterministic, if the number of zeros in the input string is even, the label is always 1. Else, it is 0.

Given the data below, determine what is the most likely label for the given string and output ONLY the label. Data:

Every zumpus is a shumpus. Polly is a lorpus. Everything that is amenable, kind, aggressive, and a grimpus is a brimpus. Each impus is a wumpus and a tumpus. Wumpuses are impuses. Everything that is floral and a shumpus is a tumpus. Yumpuses are impuses, lorpuses, and sterpuses. Max is dull. Wren is a tumpus. Everything that is transparent or a rompus is a lempus. Alex is an impus or a vumpus. Sally is temperate. Each yumpus is a zumpus. Everything that is fruity and a numpus is a grimpus. Every zumpus is not discordant. Everything that is windy or a gorpus is a vumpus. Every yumpus is a gorpus. Everything that is opaque, transparent, or a jompus is a lempus. Each rompus is a lempus. Stella is large and small and a grimpus and a gorpus. Max is a sterpus or a jompus or a gorpus. Sam is a wumpus or a dumpus or a tumpus. Everything that is bitter and a tumpus is a rompus. Everything that is angry and a lempus is a sterpus. Sally is a gorpus, a shumpus, a dumpus, or a tumpus. Everything that is snowy, sunny, and a yumpus is a lorpus. Everything that is transparent and a vumpus is a brimpus. Everything that is small and a vumpus is a rompus. Everything that is a brimpus, a shumpus, or a wumpus is a tumpus.

00001100100100100000000:
0
00100100001001110010:
0
000010010000110010:
0
00001100100100100000000:

---

Prompt 1: Prompts for description (red), word salad (blue), and modus ponens (grey) for PARITY. In the implementation these are the standard ChatML-formatted prompts, so the coloured lines are part of the system prompt, while the grey are exemplars alternating between the user input (the binary string) and the assistant's response (the single bit). In zero-shot, the prompt includes a specification of the output format ('Give your answer as a single integer, and your reasoning in a new line. For example:'). The reasoning is cut off and does not affect the experiments. The list of words from word salad comes from the ontology by Saparov & He (2023).

This is a pattern matching task. The strings in this task are generated from a probabilistic automaton. Each string is labelled 0 or 1 depending on whether the pattern "abcabb" is (1) or is not (0) in the string.
Your job is to learn what is the likelihood of a string to be labelled 0 or 1, and output the correct label.
In the limit where the automaton is deterministic, if the pattern is present in the string, the label is always 1. Else, it is 0.
Given the data below, determine what is the most likely label for the given string and output ONLY the label.
Data:

abaababbbbbbbaaaaaaaaaacabaaabcaaaaaaccbbbaaababbbbbbbbbbbbbbbbbbbbbbbbbaaaacaccbaccc
cccbcbbbaaaaccccccccaacabaaaacabaaaaccbbbbccccccaaaacabbbbaaaaaaaaaaaaaaaaaaaaccbbbaaaaaa
aabbbaabbccccccaaaccaacabbbbaabcbbccccccaaacabbbbbccccccaaaaaccbbbccccccccccabcbbacaab
aaaaaacabbaaaaaaaaaaaac:
Let's think and solve this step-by-step. We read the string character-by-character and keep a tally:
We read "a". It is a match. Our tally is: a. Now we move to the next character.
We read "b". It is a match. Our tally is: ab. Now we move to the next character.
We read "a". It is not a match. We clear our tally. Now we move to the next character.
We read "a". It is a match. Our tally is: a. Now we move to the next character.
We read "b". It is a match. Our tally is: ab. Now we move to the next character.
We read "a". It is not a match. We clear our tally. Now we move to the next character.
We read "b". It is not a match. Now we move to the next character.
...
We read "a". It is a match. Our tally is: a. Now we move to the next character.
We read "a". It is not a match. We clear our tally. Now we move to the next character.
We read "a". It is a match. Our tally is: a. Now we move to the next character.
We read "a". It is not a match. We clear our tally. Now we move to the next character.
We read "a". It is a match. Our tally is: a. Now we move to the next character.
We read "a". It is not a match. We clear our tally. Now we move to the next character.
We read "a". It is a match. Our tally is: a. Now we move to the next character.
We read "a". It is not a match. We clear our tally. Now we move to the next character.
We read "c". It is not a match. Now we move to the next character.
We have reached the end of the string. The pattern "abcabb" is is not in the string.
So the answer is 0

abaababbbbbbbaaaaaaaaaacabaaabcaaaaaaccbbbaaababbbbbbbbbbbbbbbbbbbbbbbbbaaaacaccbaccc
cccbcbbbaaaaccccccccaacabaaaacabaaaaccbbbbccccccaaaacabbbbaaaaaaaaaaaaaaaaaaaaccbbbaaaaaa
aabbbaabbccccccaaaccaacabbbbaabcbbccccccaaacabbbbbccccccaaaaaccbbbccccccccccabcbbacaab
aaaaaacabbaaaaaaaaaaaac:

Prompt 2: Prompt for one-shot Pattern Matching CoT. As before, the exemplar (grey) contains both the sample pattern and a procedurally-generated CoT response keeping a tally (stack). During SoT, both the words in the system prompt (red) and the exemplars are replaced by random words. The only things left are the pattern, labels, and the final output format.

You are helping me complete a maze. You will be given a maze almost solved, and sequence of moves to finish solving it.
Your job is to determine whether the moves are correct and will lead to solving the maze solved. You must always output 0 (incorrect) or 1 (correct).
The path you must complete is denoted by uninterrupted "+", and your completion starts at "?".
Walls are denoted by "#", and the start and end are "S" and "E", respectively.
The first move you must verify is the one connecting the path to "?".
Given the data below, determine what is the most likely label for the given maze and moves; and output ONLY the label.
Data:

Solved maze:
```
#S#######
#+++# #
# #+# ###
# #+++# #
# ###+# #
# #?++# #
# # ### #
# # ++++E
#########
```
Missing moves:
down,down,right:

Let's think and solve this step-by-step.
We begin at line 0.This line does not contain "?".
We move on then to line 1.
This line does not contain "?".
We move on then to line 2.
This line does not contain "?".
We move on then to line 3.
This line does not contain "?".
We move on then to line 4.
This line does not contain "?".
We move on then to line 5.
This line contains "?".
The "?" character is at position 3 in the line. We will now perform a search on the neighbours to find the path.
This has neighbours: ['down'] at [(6, 3)].
We select the neighbour at (6, 3) ("down") and add it to our buffer. Our buffer is: ['down'].
This has neighbours: ['down'] at [(7, 3)].
   We select the neighbour at (7, 3) ("down") and add it to our buffer. Our buffer is: ['down', 'down'].
   This one has the following available neighbours connecting to the path: ['right'] at [(7, 4)].
      This has a "+" neighbour at (7, 4) ("right"), so it connects to the path.
      We add it to our buffer. Our buffer is now ['down', 'down', 'right'].
We are done!
Our final set of positions is down,down,right and the solution says down,down,right.
So the answer is 1

Prompt 3: Prompt for the CoT version (one-shot, user input omitted) of Maze Complete. In red, the system prompt. In grey, a single exemplar.

This is a string detection task. The strings in this task are generated from a probabilistic automaton, described in the code below.

Each input is of the form LEFT#RIGHT. Each string is labelled 0 or 1 depending on whether the RIGHT pattern is (1) or is not (0) a reversal of LEFT.

Your job is to learn what is the likelihood of a string to be labelled 0 or 1, and output the correct label.

Here's the code:

ALPHABET = ["gfx", "chtte", "%", "ltintprk", "¯\\_(ツ)_/¯"]
MIN_LEN = 5

```
def reversal_tape(P):
    # n + 1 states: ALPHABET + "final"
    tape = []
    end_state = "stop"
    current_state = "start"
    states = [a for a in ALPHABET] + [end_state]
    while True:
        next_state = random.choices(states, weights=P[current_state])[0]
        if next_state == end_state:
            break
        else:
            tape.append(next_state)
        current_state = next_state

    return tape
```

Given the data below, determine what is the most likely label for the given string and output ONLY the label.
Data:

chttechttegfxltintprk%%%%%¯\\_(ツ)_/¯chtteltintprkltintprk%%#%%ltintprkltintprkchtte¯\\_(ツ)_/¯%%%%%ltintprkgfxchttechtte:

Prompt 4: System prompt (red) for the DE version (one-shot) of Stack along with a user input (grey). Remark that in this setup the alphabet is explicitly defined, as well as the code used to generate the output minus the weights for $\mathcal{P}$.

You are a vending machine. You are given a sequence of additions of balance (+10, +5, etc) or a selection (soda, biscuit, or coffee).
Your job is to output the remaining balance given the sequence.
Each soda is worth 25. Each biscuit is 20. Each coffee is 15. When someone selects a soda, biscuit, or coffee, the value of the item is subtracted from the balance.
Given the data below, determine what is the most likely balance for the given string and output ONLY the balance.
Data:

loud orange shy, rompuses amenable rompuses lorpuses Max Vumpuses or Grimpuses feisty muffled, opaque, Yumpuses rompus, gorpus angry, jompuses bitter sterpus vumpuses, shumpus luminous temperate luminous, Fae windy bitter, brown sweet, dumpuses, grimpus angry hot moderate grimpus, Rompuses lempus

+10,+10,biscuit,+10,+5,coffee,+10,+10,biscuit,+10,+10,biscuit,+5,+10,+10,soda,+10:
Let's think and solve this step-by-step. We start with a balance of 0.
impus, gorpuses "+10", Rompuses brimpus, aggressive 10 impuses Lempuses impus rainy grimpuses, sunny zumpus, shumpus 10.
spicy, bitter "+10", lorpuses floral, gorpus 10 is tumpus wumpuses numpuses slow overcast, Sterpuses vumpuses 20.
aggressive impuses "biscuit", Each Sterpuses vumpus biscuit liquid mean 20 lempus fast dumpus, fruity aggressive red, 0.
sterpus, metallic "+10", numpus, wumpuses, brimpuses, 10 windy cold zumpuses sterpus zumpuses vumpuses, happy earthy, 10.
fruity tumpuses, "+5", sterpuses, lorpuses, jompuses, 5 brown, or hot, grimpuses Gorpuses wumpus Wumpuses happy 15.
yumpuses, an "coffee", Sam windy vumpuses, coffee tumpuses dull 15 dumpuses sweet feisty Polly brown, shumpuses, 0.
...
sweet, luminous "biscuit", large moderate brimpus, biscuit dumpuses, bitter, 20 brown zumpuses snowy gorpus Everything numpuses, 0.
brimpuses discordant, "+5", rompus, melodic, jompuses, 5 wumpus sweet, Zumpuses zumpus, jompus, opaque lorpus Fae 5.
bitter brown, "+10", bright, tumpus sour, 10 yumpus, gorpus Lorpuses discordant impuses lempuses wumpus slow, 15.
gorpuses shy, "+10", opaque fruity mean 10 Stella Rompuses windy wumpus rainy, happy, jompuses dumpus, 25.
spicy Lempuses "soda", Brimpuses bitter metallic, soda brown, tumpus, 25 snowy, lorpus kind, temperate small moderate 0.
nervous, vumpuses "+10", liquid moderate impus, 10 angry snowy bright gorpuses impuses, brown, Lorpuses brown 10.

We read "+10", so we add 10 to our current balance and we now have 10.
We read "+10", so we add 10 to our current balance and we now have 20.
We read "biscuit", so we return a biscuit and substract 20 from our balance and now we have 0.
We read "+10", so we add 10 to our current balance and we now have 10.
We read "+5", so we add 5 to our current balance and we now have 15.
We read "coffee", so we return a coffee and substract 15 from our balance and now we have 0.
...
We read "biscuit", so we return a biscuit and substract 20 from our balance and now we have 0.
We read "+5", so we add 5 to our current balance and we now have 5.
We read "+10", so we add 10 to our current balance and we now have 15.
We read "+10", so we add 10 to our current balance and we now have 25.
We read "+10", so we add 10 to our current balance and we now have 10.

Our final balance is 10. The answer is then 10

Prompt 5: Prompt with one exemplar for Vending Machine (Sum). In blue, the SoT version of this prompt; and in red, the CoT version. For brevity, we omit lines for both CoT/SoT outputs. In grey are the lines shared by both prompts: the first line is the user input, and the other two are shared boilerplate for the CoT/SoT prompts for parsing. Observe how the relevant quantities do not change.

