# OpenReview forum: "Is In-Context Learning Learning?"
_ICLR.cc/2026/Conference — ICLR 2026 Poster_

### Official Review · Reviewer_ACYp · 2025-10-29

**Soundness:** 3
**Presentation:** 3
**Contribution:** 3
**Rating:** 6
**Confidence:** 3

**Summary:**

The paper argues that ICL constitutes a form of learning. It first defines ICL in the PAC-style definition of learning and then conducts an extensive empirical study to support the claim. The work contributes towards a foundational understanding of ICL, characterizing ICL as a learning paradigm rather than just prompt-based problem solving. It shows interesting results that performance gaps between different LLMs and prompts reduce as the number of exemplars grow, suggesting that ICL emerges due to autoregression paradigm. Experiments further show that ICL relies on  ad-hoc pattern extraction from exemplars, which causes brittleness for shifted distributions and inconsistency across closely-related tasks. Another interesting finding show that lexical features has limited impact on the ICL performance given enough exemplars.

**Strengths:**

1. The work presents a clean and well-controlled evaluation of ICL, first setting up in the PAC-style learning definition and then empirically testing the theory w.r.t. different prompting strategies and task designs.
2. The use of established tasks and prompt designs makes it easy to comprehend the setup. Further, the choice of formal-language based tasks provide a more principled probe to find if ICL learns underlying rules or statistical pattern matching.
3. The experimental ablations are thorough and cover a good range of controlled conditions including prompting strategies, exemplar order and distribution shifts.
4. It is an interesting insight that the exemplar ordering has limited effect, especially as the number of exemplars grow. This challenges the prior claims of success and brittleness in a few-shot setting.
5. The inclusion of traditional baselines like kNN is important, which it often overlooked by many ICL studies.
6. The paper is well written and easy to follow. The study contains all details in order to reproduce the results including prompt designs, call parameters and code.

**Weaknesses:**

The study claims that ICL emerges from a combination of prompt input and model’s intrinsic knowledge encoded in the learned weights. However, the experimental setup uses synthetic symbols with no semantic grounding. This restricts the model from leveraging its pretrained knowledge, which is a key component of how ICL functions in real-world natural language LLMs. Since the tasks in this study rely on artificial alphabets and rule-based structures, the findings may underestimate the ability of LLMs in semantically meaningful contexts, where stored knowledge plays a role during inference. Please discuss the implications of this design choice for generalizing the results to language-based ICL scenarios.

More questions and weaknesses are included in the Questions sections.

**Questions:**

1. In the introduction, the authors state that the study performs empirical studies on sensitivity to pretraining and memorization.  However, no experiments explicitly evaluate these factors. Memorization is partially controlled for through label noise, but its effects are not examined, and no experiments investigate the impact of pretraining sensitivity. Can authors please discuss these implications on ICL performance.

2. Stack and Reversal tasks require saving state representation. In the studied inference-only setup, autoregressive transformers have no explicit memory. Poor performance on PDA tasks may therefore be due to architectural constraints rather than inherent limitations of ICL. It would strengthen the paper by including an analysis on how much of the difficulty is due to transformer architecture versus the properties of ICL.

3. The study enforces strict parsing of model outputs, which I acknowledge is needed for consistent measurement. However, formatting failures are penalized the same as incorrect predictions, potentially mixing output style with task understanding. It would add transparency to the work, if the authors could clarify how much of the observed error is attributed to parsing issues rather than reasoning failures.

4. What does 1.89M datapoints/predictions comprise? Provide more details about these datapoints. Specifically, does it count only the test predictions, or are the exemplars included in each prompt also considered datapoints? A breakdown of the total by task, number of shots, and OOD settings would help readers better understand the evaluation schema.

---

> ### Author Response · Authors · 2025-11-20
>
> Thank you for your comments! We will keep it short and respond point-by-point.
>
> >The study claims that ICL emerges from a combination of prompt input and model’s intrinsic knowledge encoded in the learned weights. However, the experimental setup uses synthetic symbols with no semantic grounding. This restricts the model from leveraging its pretrained knowledge, which is a key component of how ICL functions in real-world natural language LLMs. Since the tasks in this study rely on artificial alphabets and rule-based structures, the findings may underestimate the ability of LLMs in semantically meaningful contexts, where stored knowledge plays a role during inference. Please discuss the implications of this design choice for generalizing the results to language-based ICL scenarios.
>
> This is by design: we are looking at the ICL mechanism, rather than the LLM's pretrained knowledge/linguistic abilities. The claim on intrinsic knowledge is because of the fast (yet eventually comparable) performance of the word-salad prompts versus their non-randomised counterparts.
> However!! Another reviewer also requested a discussion on the findings with respect to natural language, and we will oblige. This is a good callout which (pending some careful writing) _must_ be part of this paper.
>
> >(...) the authors state that the study performs empirical studies on sensitivity to pretraining and memorization. However, no experiments explicitly evaluate these factors. Memorization is partially controlled for through label noise, but its effects are not examined, and no experiments investigate the impact of pretraining sensitivity. Can authors please discuss these implications on ICL performance.
>
> Although memorisation is evaluated by creating wholly synthetic datasets with out-of-domain tokens (plus word salad versions), sensitivity to pretraining is not directly explored beyond what we discussed in the previous question. We will include discussions on both to make them clearer.
>
> >Stack and Reversal tasks require saving state representation. In the studied inference-only setup, autoregressive transformers have no explicit memory. Poor performance on PDA tasks may therefore be due to architectural constraints rather than inherent limitations of ICL. It would strengthen the paper by including an analysis on how much of the difficulty is due to transformer architecture versus the properties of ICL.
>
> The assumption is that autoregression functions as a memory stack, with CoT (and SoT, ideally) tracking the stack. We discuss this in L884. Indeed, certain transformer architectures are able to simulate these functions (in theory; see. e.g., a reviewer's suggestion of 'Constant Bit-Size Transformers are Turing Complete').
>
> Don't get us wrong: this is a great suggestion (transformer vs ICL). However, it is difficult to answer it, as, to the best of our knowledge, ICL has only been applied to autoregressive models (hence our scope from L53; s. 3.3) and some encoder-decoders (e.g., T5). We will add this to the limitations section, but both are *excellent* questions that the community should explore under another framing (i.e., 'is ICL intrinsic to the transformer?').
>
> >The study enforces strict parsing of model outputs, which I acknowledge is needed for consistent measurement. However, formatting failures are penalized the same as incorrect predictions, potentially mixing output style with task understanding. It would add transparency to the work, if the authors could clarify how much of the observed error is attributed to parsing issues rather than reasoning failures.
>
> This is a good point. The rationale behind it (also cut, due to space) is because it is expected that the task understanding/learning includes the signature of the output. _That said_, we examined this assumption as well and performed an ablation study on this (Compliance vs Learning; 6.4, also E.6). The gist of it is that if only one accounts for reasoning failures (ablating out compliance) the accuracy goes up. However, we also note that this makes certain phenomena, such as CoT's sensitivity to OOD, harder to spot.
>
> >What does 1.89M datapoints/predictions comprise? Provide more details about these datapoints. Specifically, does it count only the test predictions, or are the exemplars included in each prompt also considered datapoints? A breakdown of the total by task, number of shots, and OOD settings would help readers better understand the evaluation schema.
>
> It is only for the test set. Every entry in the point-wise accuracies (e.g., accuracy at 20-shot, delta=0.85) is an average of the 1,000 entries for the delta=0.85 test set.
> The exemplars do not count as predictions (they are drawn from the training set; which is always at delta=0). While there is a breakdown of the OOD setting in Appendix C and a (rushed!) description in 4.5, we do recognise this was confusing. It was a section that had to be heavily cut to fit in the allotted space. We will fix this.

---

### Official Review · Reviewer_CLXz · 2025-11-01

**Soundness:** 3
**Presentation:** 3
**Contribution:** 1
**Rating:** 6
**Confidence:** 3

**Summary:**

The paper presents a formal view of in-context learning (ICL) under a probably approximately correct(PAC)-style framework, where the prediction rule is treated as a learner conditioned on exemplars and prompt. The authors define an explicit ICL objective (Eq. 4) and argue that although ICL can be interpreted as a form of learning under this framework, its empirical behavior is largely driven by pattern matching associated with autoregressive inference rather than robust rule acquisition.

The empirical study evaluates four LLMs (two open source; two proprietary) on nine synthetic formal-language-style tasks, including FSA-type and PDA-type languages and simple arithmetic. Several prompting strategies are tested: few-shot with and without task descriptions, chain-of-thought, APO, direct token encoding, and randomized “word-salad’’ prompts.

Main empirical observations:
- Accuracy improves with additional exemplars across different model families during ICL, which shows model learn to fit exemplars; performance across models and prompt types becomes more similar at higher shot counts. Peak performance often appears around 50-100 shots, instead of at typical few-shot settings.
- Performance is stable when exemplar distributions and ordering vary within a task, but accuracy degrades under test-time distribution shifts. Chain-of-thought(CoT) and APO prompts show the strongest degradation under these shifts.
- Cross-task generalization is inconsistent. Tasks with similar underlying structure can differ by up to approximately 31 percentage points in peak accuracy. On roughly half of the tasks, simple supervised baselines (DT, kNN, MLP) achieve performance comparable to or better than ICL.

**Strengths:**

- Provides a clear PAC-style formulation of ICL, giving a precise mathematical objective (Eq. 4).
- Evaluates multiple model families (open-source and proprietary), multiple style tasks and multiple prompting strategies, reducing model-specific/task-specific bias and show conclusion can be applied generally.
- Quantifies how performance changes with the number of exemplars, showing that gains often appear at higher shot counts (50-100) and use this to decouple the influence of prompt and examplars.
- Distribution shift analysis to show the limitation of ICL

**Weaknesses:**

- The evaluated model set does not include recent models(all models were released last year). This limits the scope of the empirical claims, especially for open-source models, and it remains unclear whether the observed patterns (e.g., OOD brittleness) would hold for newer models especially there are many "reasoning" models. Although the authors mentioned "running synchronously a single task for an LLM could take over four months" but all model providers or routers support async call, which makes the claim of "synchronously" not a blocker.
- Prior work has already discussed the role of data distribution in driving ICL behavior (e.g., [1]). The paper revisits related themes, although in a different synthetic-task setting.
- The empirical coverage is restricted to single-call autoregressive prompting. The formal discussion gestures toward broader claims about ICL, but the results may not extend to multi-step or multi-call inference settings.
- Parts of the presentation could be improved: legends in Figures 2-3 are difficult to read and Figure 2 should be before Figure 3. Also there are some typos like "noting noting" in Section 3.
- The task suite focuses on formal-language classification and simple arithmetic. While this yields controlled evaluation, it is unclear how the reported effects transfer to more realistic natural-language tasks with richer structure.
- Path/graph-style tasks are evaluated without strong algorithmic baselines. This makes it harder to determine whether the model is failing due to true reasoning difficulty or due to prompt/format limitations.

[1] Chan et al., Data distributional properties drive emergent in-context learning in transformers, 2022

**Questions:**

see weaknesses section

---

> ### Author Response · Authors · 2025-11-20
>
> Thank you for your review and the comprehensive summary of the paper! Here's a few things that we would like to clarify.
>
> >The evaluated model set does not include recent models(all models were released last year). This limits the scope of the empirical claims, especially for open-source models, and it remains unclear whether the observed patterns (e.g., OOD brittleness) would hold for newer models especially there are many "reasoning" models. Although the authors mentioned "running synchronously a single task for an LLM could take over four months" but all model providers or routers support async call, which makes the claim of "synchronously" not a blocker.
>
> They do (in fact, some of these models were called async, in parallel machines to cover all tasks). The problem is that the cost is... a bit high. This study was very generously funded, although it unfortunately ended (thus why we were unable to extend this to reasoning models, e.g.). We will be more clear on this in the limitations.
>
> >Prior work has already discussed the role of data distribution in driving ICL behavior (e.g., [1]). The paper revisits related themes, although in a different synthetic-task setting.
>
> Yes! We had to cut a lot of prior work due to space. We will restore it, and add this and other works to the literature review.
>
> >The empirical coverage is restricted to single-call autoregressive prompting. The formal discussion gestures toward broader claims about ICL, but the results may not extend to multi-step or multi-call inference settings.
>
> This is correct, and specified in L214-218. We only focus on next-token prediction and ICL as a (single) learner. We will be clearer across our work on this, since this is good feedback.
>
> >Parts of the presentation could be improved: legends in Figures 2-3 are difficult to read and Figure 2 should be before Figure 3. Also there are some typos like "noting noting" in Section 3.
>
> Noted. We will go through the paper carefully to check these and any other typos as well.
>
> >The task suite focuses on formal-language classification and simple arithmetic. While this yields controlled evaluation, it is unclear how the reported effects transfer to more realistic natural-language tasks with richer structure.
>
> This is a constant feedback! To avoid repeating what we mentioned to the other reviewers: (1) we will expand on this; and (2) yes, this work solely focuses on ICL, not ICL + linguistic knowledge. However, this is definitely something that we will now highlight front and centre.
>
> >Path/graph-style tasks are evaluated without strong algorithmic baselines. This makes it harder to determine whether the model is failing due to true reasoning difficulty or due to prompt/format limitations.
>
> We did not evaluate algorithmic baselines because the standard is a (non-learning) heuristic; namely A* or similar (L271). It would be then non-comparable. However, it is known that neural networks can solve these problems as well (L820).
> That said, it is worth noting that not all path/graph problems have low performance. If anything, they have varying performances: from poor (Maze Solve) to reasonable (Maze Complete) to excellent (Hamiltonian). Given that the input format is about the same in Mazes (and almost the same prompt), we may conclude that it is not the prompt, but the problem, that is difficult.
> However, we do recognise that there are nuances here that we should discuss further (and we will add to discussion now that we have space).

---

### Official Review · Reviewer_pfQS · 2025-11-01

**Soundness:** 3
**Presentation:** 2
**Contribution:** 2
**Rating:** 4
**Confidence:** 4

**Summary:**

The paper characterizes learning as generalization to OOD examples. It evaluates the ability of ICL to perform such generalization on a range of tasks. It finds that generalization performance decreases with test examples that are increasingly OOD. It also evaluates this gap under specific conditions, e.g. with and without chain-of-thought reasoning or automated prompt optimization, or replacing natural language with nonsense words.

**Strengths:**

The paper poses an interesting question ("is in-context learning learning"), and makes an admirable attempt at trying to resolve a philosophical/theoretical question as an empirical exercise. It poses a particular formalism for the question. It does a good job at evaluating on a wide range of tasks and models.

**Weaknesses:**

I think that the main weaknesses of the paper are wrt to the framing:
* the paper assumes a specific interpretation of learning. But there are many others. E.g. a more statistical view that evaluates learning as performance on a single distribution (from which we sample both training data and test data). Or e.g. an optimization view  that sees it as error minimization or loss minimization. Some interpretations of learning (and criticisms of the term "in-context learning" even require a permanent modification to the model, e.g. via weights). It should be acknowledged that this is just one particular interpretation of what learning is.
* it is difficult to say what exactly is OOD. (a) it would be great if the paper could more explicitly describe the concrete choices for selecting the OOD distributions Q (if this is in the appendix, there should be a pointer. apologies if I missed it!). (b) I think the decision to use delta as a measure of distance actually really well supports a framing that OOD-ness is on a spectrum. This could be emphasized more. And I would recommend it not be definitely stated that "ICL performs learning"
why is the analysis limited to binary classification? this should be raised as a limitation, or otherwise it should be explained why this is more general

The clarity of the exposition can also be improved. For example:
* I think it will be especially helpful to clearly distinguish for the reader two things: generalization from pretraining data, vs generalization from the data in context (which is what's studied here). I didn't really understand what the paper was evaluating until section 4. It would be helpful if you said this earlier in the paper, and made that quite clear! E.g. something like "We evaluate the generalization of ICL by evaluating k-shot learning from context, where the query (which follows the k-shot examples) is either from the same distribution or a different distribution." (please refine/correct as needed).
* I'm confused about this statement: "That is, it modifies its states at run-time to–ideally–generalise to new observations for any task and any Q". Why should it also generalize to any task (the Valiant 1984 definition only seems to concern Q)?
* I believe that `delta` is overloaded
* section 3.1 "features are labelled with an unknown function c" -> perhaps "examples are labelled…" would be clearer. I initially misunderstood the setup as each x being just a feature of a larger example
* in the contributions section, it's unclear what "accuracy gap" you are referring to


Suggestions for missing citations:
* citations: many-shot ICL. this paper looked at many of the same questions: increasing k, and also order sensitivity
* the paper states that it takes "LLMs as recognisers of unknown expressive power" – but there's been a lot of formal work on this question. e.g. "Constant Bit-size Transformers Are Turing Complete", ""Chain of Thought Empowers Transformers to Solve Inherently Serial Problems" and many more

**Questions:**

see above for questions

---

> ### Author Response · Authors · 2025-11-20
>
> Many thanks for this review! We appreciate the time you took to read the paper and this detailed feedback (which can be summed up as: 'we will make it clearer, apologies'):
>
> >the paper assumes a specific interpretation of learning. But there are many others. E.g. a more statistical view that evaluates learning as performance on a single distribution (from which we sample both training data and test data). Or e.g. an optimization view that sees it as error minimization or loss minimization. Some interpretations of learning (and criticisms of the term "in-context learning" even require a permanent modification to the model, e.g. via weights). It should be acknowledged that this is just one particular interpretation of what learning is.
>
> Indeed. We chose the PAC learning model because it is _the_ statistical model of learning. HOWEVER, there exist other models from theoretical computer science beyond Valiant's, and another reviewer suggested we added them and discuss their implications to our findings. We wholly agree!!
>
> >(a) it would be great if the paper could more explicitly describe the concrete choices for selecting the OOD distributions Q (if this is in the appendix, there should be a pointer. apologies if I missed it!).
>
> It is in the appendix (App. C), but with the sheer amount of information and the fact that Results does not start until page 6, anyone could have missed it. We will make it clearer too.
>
> >(b) I think the decision to use delta as a measure of distance actually really well supports a framing that OOD-ness is on a spectrum. This could be emphasized more.
>
> Thank you! There is a very very very brief note in L284, but it is hard to see. That specific section underwent many cuts to make it fit into the space.
>
> >And I would recommend it not be definitely stated that "ICL performs learning"
>
> Right. Yes, this is a good recommendation. We had some concerns when writing this that it would be misinterpreted by some readers, so any suggestions on how to make the message more nuanced are very much welcome. We will note it.
>
> >why is the analysis limited to binary classification? this should be raised as a limitation, or otherwise it should be explained why this is more general
>
> Mostly because in learning theory every classification problem is a decision problem ('does this belong to a given set or not?') and k-ary classification could be extended from the same principles in linear time. That said, that (handwavy) answer is a theorist's answer and--as the paper mentions over and over--empiricism is important to determine _practical_ limits. We will call it out clearly as a limitation.
>
> >The clarity of the exposition can also be improved.
>
> Yes (and it will).
>
> >I think it will be especially helpful to clearly distinguish for the reader two things: generalization from pretraining data, vs generalization from the data in context (which is what's studied here). I didn't really understand what the paper was evaluating until section 4 (...)
>
> Yikes. This is a major problem--we will be clearer on this. Apologies, and thank you for the suggestion!
>
> >I'm confused about this statement: "That is, it modifies its states at run-time to–ideally–generalise to new observations for any task and any Q". Why should it also generalize to any task (the Valiant 1984 definition only seems to concern Q)?
>
> Yes. This is mostly an indication that an LLM is expected to act as a generalist for any task T (even though they really shouldn't be), and _also_ expected to learn properly said T (hence the 'any Q'; which could read 'for any Q for said T'). We'll rephrase it.
>
> >I believe that delta is overloaded
>
> (where? 99% sure it is, but could you point us to where?)
>
> >section 3.1 "features are labelled with an unknown function c" -> perhaps "examples are labelled…" would be clearer. I initially misunderstood the setup as each x being just a feature of a larger example
>
> Yes, needs rewriting. Thank you!
>
> >in the contributions section, it's unclear what "accuracy gap" you are referring to
>
> Will fix this too!
>
> >Suggestions for missing citations (...):
>
> We have noted them all. We had a fuller related work section but the paper suffered a considerable amount of cuts prior to submission. We will restore it and add these--much appreciated!!

---

> > ### Comment · Reviewer_pfQS · 2025-11-24
> >
> > Thank you for your response. I will await the edits.

---

### Official Review · Reviewer_8VB7 · 2025-11-03

**Soundness:** 2
**Presentation:** 2
**Contribution:** 2
**Rating:** 2
**Confidence:** 2

**Summary:**

This paper seeks to answer whether in-context learning (ICL) in large language models (LLMs) constitutes genuine learning rather than memorization or pattern recognition.

The authors start by reframing PAC-learning to define ICL formally as a process that can, in principle, satisfy generalization criteria. They then perform a large-scale empirical study (1.89 million model predictions) across four LLMs (GPT-4 Turbo, GPT-4o, Mixtral-8×7B, Phi-3.5 MoE), nine formal-language tasks, and multiple prompting strategies (description, CoT, APO, direct encoding, word-salad, etc.).

They report that:

1. Increasing the number of exemplars (shots) consistently improves accuracy, reducing model and prompt variance.

2. ICL is robust to training-distribution perturbations but brittle under test-distribution shifts, especially for CoT and APO prompts.

3. Closely related tasks show inconsistent performance gaps (up to 31%).

The authors conclude that ICL mathematically qualifies as learning but is limited by the autoregressive paradigm’s ad-hoc encoding and its sensitivity to distributional properties.

**Strengths:**

1. The study covers a large number of models, prompts, and ablations—possibly one of the most comprehensive empirical ICL analyses on synthetic tasks.

2. The paper tackles an important and widely discussed conceptual question—whether ICL counts as “learning”—and attempts to formalize this using PAC-style reasoning.

3. The inclusion of open-sourced data, prompts, and methodology is commendable and aligns with ICLR’s reproducibility standards.

**Weaknesses:**

1. The paper repeatedly asserts strong conclusions (“ICL is learning in the limit”) without solid theoretical or causal evidence linking empirical findings to the formal definition. Long sentences and vague phrasing (“it leverages statistical features from the prompt”) obscure the actual mechanism being tested.

2. The results section is dense and under-interpreted. While the experiments are numerous, the findings are mostly confirmatory (more shots → better accuracy; CoT → less robust OOD).

3. Tables 1–2 and Figures 2–4 are difficult to interpret, filled with numeric slopes and variances but lacking qualitative takeaways or error analysis. There is no principled explanation of why related tasks diverge in performance or how exemplar count interacts with task complexity.

4. The related-work section is unstructured and uncomprehensive. For example, it largely omits the extensive empirical literature on understanding how ICL operates (e.g., feature induction, model circuit analyses).

**Questions:**

1. Can you clarify what empirical behavior would falsify your claim that ICL “is learning”?

2. Could you provide simpler visualizations (accuracy vs. shots, grouped by task type) and eliminate the overly detailed slope tables?

---

> ### Author Response · Authors · 2025-11-20
>
> Thank you for your review! We really appreciate the feedback, but we were unclear in a few things so we have asked clarification questions just to be absolutely sure we are addressing your concerns.
>
> >The paper repeatedly asserts strong conclusions (“ICL is learning in the limit”) without solid theoretical or causal evidence linking empirical findings to the formal definition.
>
> Ah this is good feedback. While there is theoretical evidence linking it (e.g., L194; 'conversely, the encoded exemplars have a major contribution in the limit' have , it was not explicitly noted in the main body of the work. We can tie it back in the discussion.
> That said, it is worth noting two things:
> 1. This paper focuses strongly on empiricism over theoretical work (in line with other works, e.g. L43; Borenstein et al., 2024), using only the theoretical framework to develop the language required for the discussion (L144).
> 2. The strong conclusions need to go. This was feedback from another reviewer and we appreciate it!
>
> >Long sentences and vague phrasing (“it leverages statistical features from the prompt”) obscure the actual mechanism being tested.
>
> We are a bit unclear on this sentence. What do you mean about the 'actual mechanism being tested'? That way we can clarify it, since we are well aware that our presentation is rather dense and could use some work.
>
> >The results section is dense and under-interpreted. While the experiments are numerous, the findings are mostly confirmatory (more shots → better accuracy; CoT → less robust OOD).
>
> This is correct, although we do note that the results section are the core empirical evidence backing the findings, along with measures of statistical significance to ensure that said findings are, well, significant. There are also ablation studies to relate (and corroborate) the main findings. These all do require some level of depth, unfortunately.
>
> >There is no principled explanation of why related tasks diverge in performance or how exemplar count interacts with task complexity.
>
> This is a good question, covered in L456-465. While we may make some conjectures, this is not something that we are able to explain, beyond noting that although the tasks are mathematically similar, the way an LLM perceives them has little to do with (say) the Chomsky hierarchy.
>
> >The related-work section is unstructured and uncomprehensive. For example, it largely omits the extensive empirical literature on understanding how ICL operates (e.g., feature induction, model circuit analyses).
>
> Yes, we have cut a lot of works to keep the paper within length. Now that we have extra space, we are adding it back (it was a common point of feedback).
>
> >Can you clarify what empirical behavior would falsify your claim that ICL “is learning”?
>
> There were a few places where it was pretty clear that the score was not increasing (slope = 0) no matter which setup/shots were used. This was particularly evident in Vending Machine (Sum) (L1057), which is perhaps unsurprising due to how difficult arithmetic is for these models.
>
> >Tables 1–2 and Figures 2–4 are difficult to interpret, filled with numeric slopes and variances but lacking qualitative takeaways or error analysis (...) Could you provide simpler visualizations (accuracy vs. shots, grouped by task type) and eliminate the overly detailed slope tables?
>
> We could! Would it be possible to indicate which tables are particularly cumbersome? We do want to capture most of the information for the reader's analysis as well, so we will keep these (but move them to the appendix). Most of the qual/error is captured either in the text referring to it, or in the captions. It is short due to space, but we will expand upon it now.

---

### Official Review · Reviewer_X2GS · 2025-11-05

**Soundness:** 3
**Presentation:** 2
**Contribution:** 3
**Rating:** 6
**Confidence:** 2

**Summary:**

This paper investigates whether in-context learning in autoregressive models constitutes "learning" in the formal sense, using a variation of the PAC framework to define robustness in learning to ground discussion. They run an empirical study with 4 LLMs on nine formal, synthetic tasks to test this. The paper concludes that mathematically, ICL is a form of learning but limited in being able to learn the observed distribution in the prompt, rather than being able to generalize.

**Strengths:**

- This paper tackles a fundamental question and provides an evidence-based answer based on formal definitions
- The result that best-of-average accuracies are achieved with 50-100 examples in the limit challenges the current few-shot understanding
- The study's scale and rigor is a strength

**Weaknesses:**

- The scope of the tasks studied in the paper is rather narrow. I understand the choice to use synthetic, formal tasks to enable a rigorous analysis and minimize reliance on pretrained natural language knowledge, it would be helpful to further sign post the scope of the findings throughout the paper (though the authors do acknowledge this), and add a discussion on which/how the findings may generalize to natural language tasks
- I'd also love to see further motivation on why the paper adopted its definition of "learning", explore alternative definitions, and what that entails for the findings of the paper

**Questions:**

- In equation 4, is it necessary to rename the datapoints to e_i from x_i?
- I'm personally curious why there are such large performance gaps between the seemingly closely related tasks

---

> ### Author Response · Authors · 2025-11-20
>
> > The scope of the tasks studied in the paper is rather narrow. I understand the choice to use synthetic, formal tasks to enable a rigorous analysis and minimize reliance on pretrained natural language knowledge, it would be helpful to further sign post the scope of the findings throughout the paper (though the authors do acknowledge this), and add a discussion on which/how the findings may generalize to natural language tasks
>
> We will add a small discussion near the findings and conclusion to indicate this. Given that the relationship of our findings to natural language is complex, we agree this must be clearly delineated. Plus, we would like to be also very careful on how to phrase them to avoid any potential misinterpretation of the results.
>
> > I'd also love to see further motivation on why the paper adopted its definition of "learning", explore alternative definitions, and what that entails for the findings of the paper
>
> This is a very interesting point! It was also raised by another reviewer. We are happy to explore this further. In short, the rationale is because the PAC model is the reigning model for statistical learning theory. It is not the _only_ model (e.g., Gold's framework could also be considered a good fit here), so this is something that we are excited to include and discuss.
>
> > In equation 4, is it necessary to rename the datapoints to e_i from x_i?
>
> Not really. We mostly wanted to highlight the distinction between exemplars and the datapoint to be classified, but we may loosen that a bit now that we have more space.
>
> >I'm personally curious why there are such large performance gaps between the seemingly closely related tasks
>
> So are we! We can only make conjectures--to stay within the scope of the work. Our thinking since we found this, and until now, is that there are more fine-grained distinctions that must be made in the characterisation of these problems. We plan to make this the subject of further work.

---

> > ### Comment · Reviewer_X2GS · 2025-11-27
> >
> > Thank you for your clarifications! I look forward to seeing the additional discussion on the generalizability of findings to natural language in your revision. I will maintain my original score for now.

---

### Author Response · Authors · 2025-11-17

We would like to thank all the reviewers for their comments. They are constructive and we appreciate how evident it is that all reviewers took the time to read the work.
As we work on improvements suggested by all (in particular, clarifying the definitions of learning and readability), we might comment directly to request clarifications in some points.

---

### Author Response · Authors · 2025-11-30
**Updates summary**

The thing around leaks was a bit confusing--coming back from holidays and seeing these emails were a bit of a whiplash.
Still, we hope the reviewers remain interested in the revised version of our work, even if they are unable to comment!

The new revision is uploaded. Here's a changelog below unifying and summarising all comments:

1. **Framing**: Framing (including callouts alternate definitions of learning, next-token prediction, limitations) has been revised all across the board:
    - Statements like 'ICL is learning' have been clarified. The edits say things along the lines of 'Formally, ICL _is_ learning. Empirically, ICL is a relatively weak form of learning', (including this specifically in the conclusion).
    - The limitations have been made even clearer in the introduction, including the fact that it is constrained to non-natural language tasks.
    - Work has been tied back to the mathematical framework in the motivation _and_ the discussions.
2. **Related work**: restored and expanded, with a brief discussion on the positioning of this work in the broader literature and some of the shortcomings of previous (and this) work.
3. **Extra discussion**: extra discussion and alternate definitions of learning; plus discussion on their (probable) relationship to natural-language ICL (impact on results, relation to other frameworks, including human models of learning, etc).
    - We have included a minor set of experiments to back the claims, with an **untrained model**.
    - In particular, noting that natural language is somewhere in the Chomsky hierarchy that is slightly more complex than it is known from this paper and other (more theoretical) results; yet the empirical performance appears to contradict (again) theoretical work, thus opening new areas of research.
    - Since **this was an area reviewers** (and us) **were interested on**, the sections are **Appendices C and D** (the latter appendix including the natural language discussions).
4. **Overall presentation**: thanks to the page limit, the text is leaner.
    - Images in the main body of work are more readable (i.e., larger text). Unfortunately, we were unable to move Fig. 3 before Fig. 2 given its size (a full page), but pointed the reader to it.
    - Typos addressed.
    - Sentences which we deemed vague have been 'concretised'
    - We reverted $e_i$ to $x_i$
    - Improved findings, discussion, conclusion, and data-generation section's readability.

---

### Meta-Review · Area_Chair_DxbW · 2026-01-18

**Summary:**

This paper asks a basic but important question: whether in-context learning (ICL) should really be viewed as learning. Reviewers generally agreed that the experimental setup, based on controlled synthetic tasks, is careful and technically solid. Most of the discussion focused on framing rather than experimental flaws. In particular, reviewers felt that some claims were initially phrased too strongly, that the PAC-style notion of learning needed clearer justification, and that it was not obvious how far results from synthetic tasks should carry over to natural-language ICL.

**Reviewer Concerns:**

Addressed by the rebuttal
- The authors toned down strong claims and clearly separated formal definitions from empirical findings, with limitations stated more explicitly.
- The choice of the PAC learning framework is now better explained and positioned among other possible definitions of learning.
- The related work and overall presentation were substantially improved.

Remaining limitations
- The experiments still focus on synthetic tasks and single-step ICL, so conclusions about natural-language or newer reasoning models remain limited.
- Some observed effects (such as large gaps between similar tasks and sensitivity to distribution shift) are not fully explained.

**Reviewer Scores:**

- Reviewer X2GS (score: 6): Unchanged (explicitly stated).
- Reviewer 8VB7 (score: 2): Likely slightly increased since clarity concerns were addressed.
- Reviewer pfQS (score: 4): Likely increased, as most framing and clarity concerns were directly addressed.
- Reviewer CLXz (score: 6): Likely unchanged or slightly higher.
- Reviewer ACYp (score: 6): Likely unchanged or slightly higher.

---

### Decision · Program_Chairs · 2026-01-26

Accept (Poster)